# STRIC: STACKED RESIDUALS OF INTERPRETABLE COMPONENTS FOR TIME SERIES ANOMALY DETECTION

## ABSTRACT

We present a residual-style architecture for interpretable forecasting and anomaly detection in multivariate time series. Our architecture is composed of stacked residual blocks designed to separate components of the signal such as trends, seasonality, and linear dynamics. These are followed by a Temporal Convolutional Network (TCN) that can freely model the remaining components and can aggregate global statistics from different time series as context for the local predictions of each time series. The architecture can be trained end-to-end and automatically adapts to the time scale of the signals. After modeling the signals, we use an anomaly detection system based on the classic CUMSUM algorithm and a variational approximation of the $f$-divergence to detect both isolated point anomalies and change-points in statistics of the signals. Our method outperforms state-of-the-art robust statistical methods on typical time series benchmarks where deep networks usually underperform. To further illustrate the general applicability of our method, we show that it can be successfully employed on complex data such as text embeddings of newspaper articles.

## 1 INTRODUCTION

Time series data is being generated in increasing volumes from industrial, medical, commercial and scientific applications. Such growth is fueling demand for anomaly detection algorithms that are general enough to be applicable across domains, yet reliable enough to operate on real-world time series data (Munir et al., 2019; Geiger et al., 2020; Su et al., 2019). While recent developments have focused on deep neural networks (DNNs), simple linear models still outperform DNNs in applications that require robustness (Braei & Wagner, 2020) and interpretable failure modes (Geiger et al., 2020; Su et al., 2019).

To harvest the flexibility and interpretability of engineered modules while enabling end-to-end differentiable training, we introduce STRIC: Stacked Residuals of Interpretable Components. We follow standard practice and consider a two stage anomaly detection pipeline comprising a model of the normal time series and an anomaly detector based on the prediction residuals (Munir et al., 2019). In particular, STRIC is composed of three modules: An interpretable *local predictor*, a *non-linear predictor* and a novel non-parametric *anomaly detector*.

More specifically, STRIC uses a parametric model implemented by a sequence of residuals blocks with each layer capturing the prediction residual of previous layers. The first layer models trends, the second layer models quasi-periodicity/seasonality at multiple temporal scales, the third layer is a general linear predictor, and the last is a general non-linear model in the form of a Temporal Convolution Network (TCN). While the first three layers are *local* (i.e. applied to each component of the time series separately), the last integrates *global* statistics from additional time series (covariates). Thanks to the residual structure the interpretable *linear* blocks do not reduce the representative power of our non-linear architecture: any component of the input time-series which cannot be modeled by the interpretable blocks is processed deeper into our architecture by the non-linear module (see Section 4). The model is trained end-to-end with a prediction loss and we automatically select its complexity using an upper bound of the marginal likelihood which, to the best of our knowledge, has never been applied to TCNs before.

Anomalies are detected by checking for time instants in which the prediction residual is not stationary. To avoid any unrealistic assumption on the prediction residuals distribution, we use a likelihood ratio test that we derive from a variational upper bound of $f$-divergences and that can be computed directly from the data points.

To summarize, our contributions are:

1. We introduce STRIC, a stacked residual model that explicitly isolates interpretable factors such as slow trends, quasi-periodicity, and linearly predictable statistics (Oreshkin et al., 2019; Cleveland et al., 1990), and incorporates statistics from other time series as context/side information.

2. We introduce a novel regularization that is added to the prediction loss and is used for automatic model complexity selection according to the Empirical Bayes framework (Rasmussen & Williams, 2006).

3. We introduce a non-parametric extesion of the CUMSUM algorithm which entails a tunable parameter corresponding to the length of observation and enables anomaly detection in the absence of knowledge about the pre- and post-distributions.

4. We test our method on standard anomaly detection benchamrks and show it merges both the advantages of simple and interpretable linear models and the flexibility of non-linear ones while discounting their major drawbacks: lack of flexibility of linear models and lack of interpretability and overfitting of non-linear ones.

## 2 RELATED WORK

A *time series* is an ordered sequence of data points. We focus on discrete and regularly spaced time indices, and thus we do not include literature specific to asynchronous time processes in our review. Different methods for time series anomaly detection (TSAD) can be taxonomized by their choice of (i) discriminant function, (ii) continuity criterion, and (iii) optimization method to determine the tolerance threshold. It is common to use statistics of the prediction error as the discriminant (Braei & Wagner, 2020), and the likelihood ratio between the distribution of the prediction error before and after a given time instant as the continuity criterion (Yashchin, 1993). Recent methods compute the discriminant using deep neural network architectures and euclidean distance as continuity criterion (Munir et al., 2019; Geiger et al., 2020; Su et al., 2019; Bashar & Nayak, 2020).

Our method follows a similar line but introduces novel elements both in (i) and (ii): (i) the discriminant function is the prediction error residual of a novel regularized stacked residual architecture; (ii) the decision function is based on a novel non-parametric extension of the CUMSUM algorithm. The resulting method, STRIC, has the advantage of separating interpretable components due to trends and seasonality, without reducing the representative power of our architecture. At initialization, the system is approximately equivalent to a multi-scale SARIMA model (Adhikari & Agrawal, 2013), which can be reliably applied out-of-the-box on most time series. However, as more data is acquired, any part of the system can be further fine-tuned in an unsupervised end-to-end fashion.

Munir et al. (2019) argue that anomaly detection can be solved by exploiting a flexible model provided a proper inductive bias is introduced (e.g. TCN). In Appendix A.7.2 we show that TCN alone might overfit simple time series. We therefore take their direction a step further, and, differently from previous works (Bai et al., 2018; Munir et al., 2019; Sen et al., 2019; Geiger et al., 2020), we provide our temporal model with an interpretable structure, similar to Oreshkin et al. (2019). Moreover, unlike previous works on interpretability of DNNs (Tsang et al., 2018; Guen et al., 2020), our architecture explicitly imposes both an inductive bias and a regularization which are designed to expose the user both a STL-like decomposition (Cleveland et al., 1990) and the relevant time scale of the signals. Since TCNs tend to overfit if not properly regularized (Appendix A.7.2), we constrain our TCN's representational power by enforcing fading memory (Zancato & Chiuso, 2021) while retaining what is needed to predict future values.

Our method outperforms both classical statistical methods (Braei & Wagner, 2020) and deep networks (Munir et al., 2019; Geiger et al., 2020; Su et al., 2019; Bergman & Hoshen, 2020; Bashar & Nayak, 2020) on different anomaly detection benchmarks (Laptev & Amizadeh, 2020; Lavin & Ahmad, 2015) (Section 6). Moreover, we show it can be employed to detect anomalous patterns on complex data such as text embeddings of newspaper articles (Figure 4).

## 3 NOTATION

We denote vectors with lower case and matrices with upper case. In particular $y$ is multi-variate time series $\{y(t)\}_{t \in \mathbb{Z}}$, $y(t) \in \mathbb{R}^n$; we stack observations from time $t$ to $t + k - 1$ and denote the resulting matrix as $Y_t^{t+k-1} := [y(t), y(t+1), ..., y(t+k-1)] \in \mathbb{R}^{n \times k}$. The row index refers to the dimension of the time series while the column index refers to the temporal dimension. We denote the $i$-th component of the time series $y$ as $y_i$ and its evaluation at time $t$ ad as $y_i(t) \in \mathbb{R}$. We refer to $\{y(s), \ s > t\}$ as the test/future and to $\{y(s), \ s \le t\}$ as the reference/past intervals. At time $t$, sub-sequences containing the $n_p$ past samples up to time $t - n_p + 1$ are given by $Y_{t-n_p+1}^t$ (note that we include the

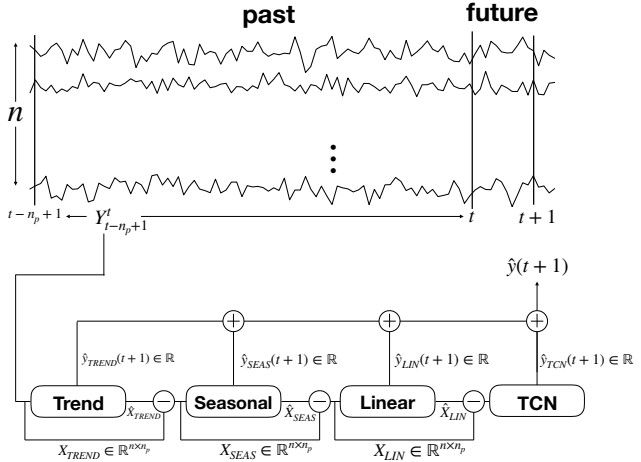

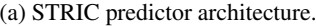

(a) STRIC predictor architecture.

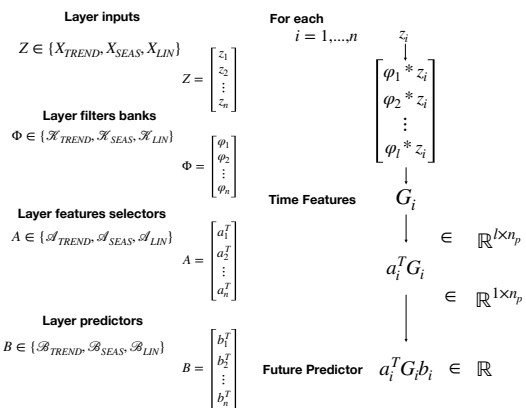

(b) **Interpretable blocks structure**. Time features are extracted independently for each time series (see Appendix A.1 for more details).

present data into the past data), while future samples up to time $t + n_f$ are $Y_{t+1}^{t+n_f}$. We will use past data to predict future ones, where the length of past and future intervals is an hyper-parameter that is up to the user to design.

## 4 Temporal Residual Architecture

Our architecture is depicted in Figure 1a. Its basic building blocks are causal convolutions (Bai et al., 2018), with a fixed-size 1-D kernel with input elements from time $t$ and earlier. Rather than initializing the convolutional filter randomly, as commonly done in deep learning, we initialize the weights so that each layer is biased to attend at different components of the signal, as explained in the following.

**Linear module.** The first (*linear*) module is interpretable and captures local statistics of a given time series by means of a cascade of learnable linear filters. Its first layer models and removes slow-varying components in the input data. We initialize the filters to mimic a causal Hodrick Prescott (HP) filter (Ravn & Uhlig, 2002). The second layer models and removes periodic components: it is initialized to have a periodic impulse response. Finally, the third layer implements a linear stationary filter bank.

We treat the impulse responses parameters of the linear filters as trainable parameters. To allow our model to work on a wide variety of time scales, we initialize the trend layer with different HP smoothness degrees, while we initialize the periodic and linear-stationary layers with randomly chosen poles (Farahmand et al., 2017) both on the unit circle and within the unit circle, thus allowing to capture different periodicities.

**Non-linear module.** The second (*non-linear*) module aggregates global statistics from different time series using a TCN model (Sen et al., 2019). It takes as input the prediction residual of the linear module and outputs a matrix $G(Y_{t-n_p+1}^t) \in \mathbb{R}^{l \times n_p}$ where $l$ is the number of output features extracted by the TCN model. The column $G(Y_{t-n_p+1}^t)_j$ with $j = 1, ..., n_p$ of the non-linear features is computed using data up to time $t - n_p + j$ (due to the internal structure of a TCN network (Bai et al., 2018)). We build a linear predictor on top of $G(Y_{t-n_p+1}^t)$ for each single time series independenty: the predictor for the $i$-th time series is given by: $\hat{y}_{\text{TCN}}(t + 1)_i := a_i^T G(Y_{t-n_p+1}^t) b_i$ where $a_i \in \mathbb{R}^l$ and $b_i \in \mathbb{R}^{n_p}$. Since $a_i$ combines features (uniformly in time) we can interpret it as a feature selector. While $b_i$ aggregates relevant features across time indices to build the one-step ahead predictor (see Appendix A.1).

Note that the third layer of the linear module is a superset of preceding ones, and the non-linear module is a superset of the whole linear module. While this makes the model redundant, we show that this design, coupled with proper initialization and regularization, improves the reliability and intepretability of the final model. We improve filters optimization by sharing their kernel parameters among different time series so that global information (e.g., common trend shapes, periodicities) can be extracted. In Appendix A.1, we describe each component of the model in detail.

## 4.1 Automatic Complexity Determination

Consider the TCN-based future predictor be $\hat{y}_{\text{TCN}}(t+1) := a^T G(Y_{t-n_p+1}^t) b = \hat{X}_{\text{TCN}} b$ where $\hat{X}_{\text{TCN}} \in \mathbb{R}^{1 \times n_p}$ is the output of the TCN block which depends on the the past window $Y_{t-n_p+1}^t$ of length $n_p$ (the memory of the predictor).[1] Ideally, $n_p$ should be large enough to capture the "true" memory of the time series, but should not be too large if not necessary (i.e. bias variance trade-off). In practice however, flexible feature extractors (such as TCNs or plain DNNs) are prone to overfitting. Therefore, some regularization is needed to control model complexity and benefit from having a large memory window. In this section, we introduce a novel regularized loss inspired by Bayesian arguments which allows us to use an architecture with a "large enough" past horizon $n_p$ (i.e., larger than the true system memory) and automatically select the *relevant past* to avoid overfitting. Such information is exposed to the user through an interpretable parameter $\lambda$ that directly measures the *relevant time scale* of the signal.

**Bayesian learning formulation**: We model the innovations (optimal prediction errors) as Gaussian, so that $y(t+1) \mid Y_{t-n_p+1}^t \sim \mathcal{N}(F^*(Y_{t-n_p+1}^t)), \eta^2)$ where $F^*$ is the optimal predictor of the future values given the past. Note that this modeling assumption does not restrict our framework and is used only to justify the use of the squared loss to learn the regression function of the predictor. In practice, we do not know $F^*$ and we approximate it with our parametric model. For ease of exposition, we group all the architecture parameters except $b$ in $W$ (linear filters parameters, TCN kernel parameters etc.) and write the conditional likelihood of the future given the past data of our parametric model as $p(Y_{t+1}^{t+n_f} \mid b, W, Y_{t-n_p+1}^t) = \prod_{k=1}^{n_f} p(y(t+k) \mid b, W, Y_{t+k-n_p}^{t+k-1})$.

To make the notation simpler, we shall denote with $Y_f := Y_{t+1}^{t+n_f} \in \mathbb{R}^{n_f}$ the set of future outputs over which the predictor is computed and we shall use $\hat{Y}_{b,W} \in \mathbb{R}^{n_f}$ as the predictor's outputs.

In a Bayesian framework, the optimal set of parameters can be found maximizing the posterior $p(b, W \mid Y_f)$ over the model parameters. We model $b$ and $W$ as independent random variables:

$$p(b, W \mid Y_f) \propto p(Y_f \mid b, W) p(b) p(W) \tag{1}$$

where $p(b)$ is the prior associated to the predictor coefficients and $p(W)$ is the prior on the remaining parameters. The prior $p(b)$ should encode our belief that the prediction model should not be too complex and should depend only on the *most relevant past*. We model this by assuming that the components of $b$ have zero mean and exponentially decaying variances: $\mathbb{E} b_{n_p-j-1}^2 = \kappa \lambda^j$ for $j = 0, ..., n_p - 1$, where $\kappa \in \mathbb{R}^+$ and $\lambda \in (0, 1)$. The corresponding maximum entropy prior $p_{\lambda,\kappa}(b)$ (Cover & Thomas, 1991) under such constraints is $\log(p_{\lambda,\kappa}(b)) \propto -\|b\|_{\Lambda^{-1}}^2 - \log(|\Lambda|)$ where $\Lambda \in \mathbb{R}^{n_p}$ is a diagonal matrix with elements $\Lambda_{j,j} = \kappa \lambda^j$ with $j = 0, ..., n_p$. Here, $\lambda$ represents how fast the output of the predictor "forgets" the past. Therefore, $\lambda$ regulates the complexity of the predictor: the smaller $\lambda$, the lower the complexity.

In practice, $\lambda$ has to be estimated from the data. One would be tempted to estimate jointly $b, W, \lambda, \kappa$ (and possibly $\eta$) by minimizing the negative log of the joint posterior (see Appendix A.2.1). Unfortunately, this leads to a degeneracy since the joint negative log posterior goes to $-\infty$ when $\lambda \to 0$. Indeed, typically the parameters describing the prior (such as $\lambda$) are estimated by maximizing the marginal likelihood, i.e., the likelihood of the data once the parameters $(b, W)$ have been integrated out. Since computing (or even approximating) the marginal likelihood in this setup is prohibitive, we now introduce a variational upper bound to the marginal likelihood which is easier to estimate.

**Variational upper bound to the marginal likelihood**: The model structure we consider is linear in $b$ and we can therefore stack the predictions of each available time index $t$ to get the following linear predictor on the whole future data available: $\hat{Y}_{b,W} = F_W b$ where $F \in \mathbb{R}^{n_f \times n_p}$ is obtained by stacking $\hat{X}_{\text{TCN}}(Y_{i-n_p+1}^i)$ for $i = t, ..., t + n_f - 1$.

**Proposition 4.1.** *Consider a model on the form:* $\hat{Y}_{b,W} = F_W b$ *(linear in $b$ and possibly non-linear in $W$) and its posterior in Equation* (1). *Assume the prior on the parameters $b$ is given by the maximum entropy prior and $W$ is fixed. Then the following is an upper bound on the marginal likelihood associated to the posterior in Equation* (1) *with marginalization taken* only *w.r.t. $b$:*

$$\mathcal{U}_{b,W,\Lambda} = \frac{1}{\eta^2} \left\| Y_f - \hat{Y}_{b,W} \right\|^2 + b^\top \Lambda^{-1} b + \log \det(F_W \Lambda F_W^\top + \eta^2 I). \tag{2}$$

This (proved in Appendix A.2) provides an alternative loss function to the negative log posterior which does not suffer of the degeneracy alluded above while optimizing over $b$, $W$, $\lambda$ and $\kappa$.

---

[1]For simplicity we consider scalar time series, but our approach easily generalizes to multivariate time series (Appendix A.2.2).

**Remark**: We use batch normalization (Ioffe & Szegedy, 2015) to normalize the output of $F_W$ along its rows so that features have comparable scales, this avoids the TCN network to counter the fading regularization by increasing its output scales (see Appendix A.2.3).

# 5 ANOMALY DETECTOR

In this section, we present our anomaly detection method based on a variational approximation of the likelihood ratio between two windows of model residuals. Our temporal residual architecture model produces the prediction residual after removing trends, periodicity, and stationary (linear) components, as well as considering global covariates. Such a prediction residual is used to test the hypothesis that the time instant $t$ is anomalous by comparing its statistics before $t$ on temporal windows of length $n_p$ and $n_f$. The detector is based on the likelihood ratios aggregated sequentially using the classical CUMSUM algorithm (Page, 1954; Yashchin, 1993). CUMSUM, however, requires knowledge of the distributions, which we do not have. Unfortunately, the problem of estimating the densities is hard (Vapnik, 1998) and generally intractable for high-dimensional time series (Liu et al., 2012). We circumvent this problem by directly estimating the likelihood ratio with a variational characterization of $f$-divergences (Nguyen et al., 2010) which involves solving a convex risk minimization problem in closed form.

In Section 5.1, we summarize the standard material necessary to derive our new estimator and the resulting anomaly test. The overall method is entirely unsupervised, and users can tune the scale parameter (corresponding to the window of observation when computing the likelihood ratios) and the coefficient of CUMSUM (depending on the application and desired operating point in the tradeoff between missed detection and false alarms).

## 5.1 LIKELIHOOD RATIOS AND CUMSUM

CUMSUM (Page, 1954) is a classical Sequential Probability Ratio Test (SPRT) (Basseville & Nikiforov, 1993; Liu et al., 2012) of the null hypothesis $H_0$ that the data after the given time $c$ comes from the same distribution as before, against the alternative hypothesis $H_c$ that the distribution is different. We denote the distribution before $c$ as $p_p$ and the distribution after the anomaly at time $c$ as $p_f$.

If the density functions $p_p$ and $p_f$ were known (we shall relax this assumption later), the optimal statistic to decide whether a datum $y(i)$ is more likely to come from one or the other is the likelihood ratio $s(y(i))$. According to the Neyman-Pearson lemma, $H_0$ is accepted if the likelihood ratio $s(y(i))$ is less than a threshold chosen by the operator, otherwise $H_c$ is chosen. In our case, the competing hypotheses are $H_0$ = "no anomaly has happened" and $H_c$ = "an anomaly happened at time $c$". We denote with $p_{H_0}$ and $p_{H_c}$ the p.d.f.s under $H_0$ and $H_c$ so that: $p_{H_0}(Y_1^K) = p_p(Y_1^K)$ and $p_{H_c}(Y_1^{c-1}) = p_p(Y_1^{c-1})$, $p_{H_c}(Y_c^K \mid Y_1^{c-1}) = p_f(Y_c^K \mid Y_1^{c-1})$. Therefore the likelihood ratio is:

$$\Omega_c^t := \frac{p_{H_c}(Y_1^t)}{p_{H_0}(Y_1^t)} = \frac{p_p(Y_1^{c-1})p_f(Y_c^t \mid Y_1^{c-1})}{p_p(Y_1^t)} = \frac{p_f(Y_c^t \mid Y_1^{c-1})}{p_p(Y_c^t \mid Y_1^{c-1})} = \prod_{i=c}^{t} \frac{p_f(y(i) \mid Y_1^{i-1})}{p_p(y(i) \mid Y_1^{i-1})} \tag{3}$$

To determine the presence of an anomaly, we can compute the cumulative sum $S_c^t := \log \Omega_c^t$ of the (log) likelihood ratios, which depends on the time $c$, and estimate $c^*$ using a maximum likelihood criterion, corresponding to the detection function $h_t = \max_{1 \le c \le t} S_c^t$. The first instant at which we can confidently assess the presence of a change point (a.k.a. stopping time) is: $c_{\text{stop}} = \min\{t : h_t \ge \tau\}$ where $\tau$ is a design parameter that modulates the sensitivity of the detector depending on the application. The final estimate $\hat{c}$ of the true change point $c^*$ after the detection $c_{\text{stop}}$ is simply given by the timestamp $c$ at which the maximum of $h_t = \max_{1 \le c \le t} S_c^t$ is achieved. In Appendix A.3, we provide an alternative derivation that shows that CUMSUM is a comparison of the test statistic with an adaptive threshold that keeps complete memory of past ratios. The next step is to relax the assumption of known densities, which we bypass in the next section by directly approximating the likelihood ratios to compute the cumulative sum.

### 5.1.1 LIKELIHOOD RATIO ESTIMATION WITH PEARSON DIVERGENCE

The goal of this section is to tackle the problem of estimating the likelihood ratio of two general distributions $p_p$ and $p_f$ given samples. To do so, we leverage a variational approximation of $f$-divergences (Nguyen et al., 2010) whose optimal solution is directly connected to the likelihood ratio. For different choices of divergence function, different estimators of the likelihood ratio can be built. We focus on a particular divergence choice, the Pearson divergence, since it provides a closed form estimate of the likelihood ratio (see Appendix A.4).

**Proposition 5.1.** *(Nguyen et al., 2010; Liu et al., 2012) Let $\phi := p_f/p_p$ be the likelihood ratio of the unknown distributions $p_f$ and $p_p$. Let $\mathcal{F} := \{f_i : f_i \sim p_f, i = 1, ..., n_f\}$ and $\mathcal{H} := \{h_i : h_i \sim p_p, i = 1, ..., n_p\}$ be two sets containing $n_f$ and $n_p$ samples i.i.d. from $p_f$ and $p_p$ respectively. An empirical estimator $\hat{\phi}$ of the likelihood ratio $\phi$ is given by the solution to the following convex optimization problem:*

$$\hat{\phi} = \arg\min_{\phi} \frac{1}{2n_p} \sum_{i=1}^{n_p} \phi(h_i)^2 - \frac{1}{n_f} \sum_{i=1}^{n_f} \phi(f_i) \tag{4}$$

**Proposition 5.2.** *(Liu et al., 2012; Kanamori et al., 2009) Let $\phi$ in Equation (4) be chosen in the family of Reproducing Kernel Hilbert Space (RKHS) functions $\Phi$ induced by the kernel $k$. Let the kernel sections be centered on the set of data $\mathcal{S}_{tr} := \{\mathcal{F}, \mathcal{H}\}$ and let the kernel matrices evaluated on the data from $p_f$ and $p_p$ be $K_f := K(\mathcal{F}, \mathcal{S}_{tr})$ and $K_p := K(\mathcal{H}, \mathcal{S}_{tr})$. The optimal regularized empirical likelihood ratio estimator on a new datum $e$ is given by:*

$$\hat{\phi}(e) = \frac{n_p}{n_f} K(e, \mathcal{S}_{tr}) \Big(K_p^T K_p + n_p \gamma I_{n_p+n_f}\Big)^{-1} K_f^T \mathbb{1}. \tag{5}$$

**Remark**: The estimator in Equation (5) is not constrained to be positive. Nonetheless, the positivity constraints can be enforced. In this case, the closed form solution is no longer valid but the problem remains convex.

## 5.2 SUBSPACE LIKELIHOOD RATIO ESTIMATION AND CUMSUM

In this section, we present our anomaly detector estimator. We test for an anomaly in the data $Y_1^t$ by looking at the prediction residuals $E_1^t$, which provide a sufficient representation of $Y_1^t$ (see Appendix A.5). We therefore assume we are given the prediction errors $E_1^t$ obtained from our time series predictor (the predictor should model the normal behaviour). This guarantees that the sequence $E_1^t$ is white in each of its normal subsequences. On the other hand, if the model is applied to a data subsequence which contains the abnormal condition, the residuals are correlated.

We estimate the likelihood ratio of $p_f$ and $p_p$ on a datum $e_t$ as $\hat{\phi}_t(e_t)$. $\hat{\phi}_t$ is obtained by applying Equation (5) on the past window of size $n_p + n_f$. At each time instant $t$, we compute the necessary kernel matrices as $K_f(E_{t-n_f+1}^t, E_{t-n_p-n_f+1}^t)$ and $K_p(E_{t-n_p-n_f+1}^{t-n_f}, E_{t-n_p-n_f+1}^t)$.

**Remark**: At time $t$, the likelihood ratio is estimated assuming i.i.d. data. This assumption holds if no anomaly happened but does not hold in the abnormal situation since residuals are not i.i.d. In Appendix A.5, we prove that treating correlated variables as uncorrelated provides a lower bound on the actual cumulative sum of likelihood ratios. For a fixed threshold, this means the detector cumulates less and therefore requires more time to reach the threshold.

Finally, we compute the detector function by aggregating the estimated likelihood ratios: $\hat{S}_c^t := \sum_{i=c}^t \log \hat{\phi}_i(e_i)$.

**Remark**: The choice of the windows length ($n_p$ and $n_f$) is fundamental and highly influences the likelihood estimator. Using small windows makes the detector highly sensible to point outliers, while larger windows are better suited to estimate sequential outliers (see Appendix A.5).

## 6 EXPERIMENTAL RESULTS

In this section, we show STRIC can be successfully applied to detect anomalous behaviours on different anomaly detection benchmarks. In particular, we test our novel residual temporal structure, the automatic complexity regularization and the anomaly detector on the following datasets: Yahoo (Laptev & Amizadeh, 2020), NAB (Lavin & Ahmad, 2015), CO2 Dataset (see Appendix A.6). In addition, to show the general applicability and flexibility of our method, we test STRIC on the challenging task of detecting anomalous events in time series generated from embeddings of articles from the New York Times. See Appendix A.7 for the experimental setup and data normalization.

**Anomaly detection**: While recent works show deep learning models are not well suited to solve AD on standard anomaly detection benchmarks (Braei & Wagner, 2020), we prove deep models can be effective provided they are used with a proper inductive bias and regularization. In Table 1, we compare STRIC against statistical and deep learning based anomaly detection methods. Our experiments follow the experimental setup and evaluation criteria used in Braei & Wagner (2020) and Munir et al. (2019). Note no other method performs consistently (across different datasets) as good as STRIC. In particular, STRIC achieves the best F1 score on Yahoo A3, in Appendix A.7 we show this is mainly

Table 1: **Comparison with SOTA anomaly detectors:** We compare STRIC with other anomaly detection methods (see Appendix A.8) on the experimental setup and the same evaluation metrics proposed in (Braei & Wagner, 2020; Munir et al., 2019). The baseline models are: MA, ARIMA, LOF (Shen et al., 2020), LSTM (Braei & Wagner, 2020; Munir et al., 2019), Wavenet (Braei & Wagner, 2020) , Yahoo EGADS (Munir et al., 2019) , GOAD (Bergman & Hoshen, 2020), OmniAnomaly (Su et al., 2019), Twitter AD (Munir et al., 2019), TanoGAN (Bashar & Nayak, 2020), TadGAN (Geiger et al., 2020) , DeepAR (Flunkert et al., 2017) and DeepAnT (Munir et al., 2019) . STRIC outperforms most of the other methods based on statistical models and based on DNNs. See Table 6 for the same table obtained by looking at the relative performance w.r.t. STRIC.

| F1-score | Yahoo A1 | Yahoo A2 | Yahoo A3 | Yahoo A4 | NAB Tweets | NAB Traffic |
|---|---|---|---|---|---|---|
| ARIMA | 0.35 | 0.83 | 0.81 | **0.70** | 0.57 | 0.57 |
| LSTM | 0.44 | 0.97 | 0.72 | 0.59 | | |
| Yahoo EGADS | 0.47 | 0.58 | 0.48 | 0.29 | | |
| OmniAnomaly | 0.47 | 0.95 | 0.80 | 0.64 | 0.69 | 0.70 |
| Twitter AD | **0.48** | 0 | 0.26 | 0.31 | | |
| TanoGAN | 0.41 | 0.86 | 0.59 | 0.63 | 0.54 | 0.51 |
| TadGAN | 0.40 | 0.87 | 0.68 | 0.60 | 0.61 | 0.49 |
| DeepAR | 0.27 | 0.93 | 0.47 | 0.45 | 0.54 | 0.60 |
| DeepAnT | 0.46 | 0.94 | 0.87 | 0.68 | | |
| STRIC (ours) | **0.48** | **0.98** | **0.89** | 0.68 | **0.71** | **0.73** |

| AUC | Yahoo A1 | Yahoo A2 | Yahoo A3 | Yahoo A4 | NAB Tweets | NAB Traffic |
|---|---|---|---|---|---|---|
| MA | 0.868 | 0.994 | 0.994 | **0.986** | | |
| ARIMA | 0.873 | 0.989 | 0.990 | 0.971 | | |
| LOF | 0.904 | 0.901 | 0.641 | 0.640 | 0.491 | 0.428 |
| Wavenet | 0.824 | 0.761 | 0.580 | 0.592 | | |
| LSTM | 0.812 | 0.735 | 0.578 | 0.589 | | |
| GOAD | 0.893 | 0.921 | 0.888 | 0.866 | 0.572 | 0.641 |
| DeepAnT | 0.898 | 0.961 | 0.928 | 0.860 | 0.554 | 0.637 |
| STRIC (ours) | **0.931** | **0.999** | **0.999** | 0.935 | **0.658** | **0.685** |

due to STRIC's predictor. In fact, most of the time series in Yahoo A3 are characterized by trend components and seasonalities which STRIC's interpretable predictor can easily model (see Appendix A.7.2). In Appendix A.7, we show some ablation studies on the effects of STRIC's hyper-parameters on its performance. In particular, we find that STRIC is highly affected by the choice of the length of the windows used to estimate the likelihood ratio, while not being much sensitive to the choice of the memory of the predictor (Appendix A.7.1). Interestingly, STRIC does not achieve the optimal F1/AUC compared to linear models on Yahoo A4. The ability of linear models to outperform non-linear ones on Yahoo A4 is known in the literature (e.g. in Geiger et al. (2020) any non-linear model is outperformed by AR/MA models of the proper complexity). The main motivation for this is that modern (non-linear) methods tend to overfit on Yahho A4 and therefore generalization is usually low. Instead, thanks to fading regularization and model architecture, STRIC does not exhibit overfitting despite having larger complexity than SOTA linear models used in A4. To conclude, we believe that STRIC merges both the advantages of simple and interpretable linear models and the flexibility of non-linear ones while discounting their major drawbacks: lack of flexibility of linear models and lack of interpretability and overfitting of non-linear ones (see Appendix A.8 for a more in depth discussion).

**STRIC interpretable time series decomposition**: In Figure 2, we show STRIC's interpretable decomposition. We report predicted signals (first row), estimated trends (second row) and seasonalities (third row) for different datasets. For all experiments, we plot both training data (first $40\%$ of each time series) and test data. Note the interpretable components of STRIC generalize outside the training data, thus making STRIC work well on non-stationary time series (e.g. where the trend component is non negligible and typical non linear models overfit, see Appendix A.7.2).

**Ablation study**: We now compare the prediction performance of a general TCN model with our STRIC method in which we remove the interpretable module and the fading regularization one at the time. In Table 2, we report the test RMSE prediction errors and the RMSE generalization gap (i.e. difference between test and training RMSE prediction errors) for different datasets while keeping all the training parameters the same (e.g. training epochs, learning rates etc.) and model parameters (e.g. $n_b = 100$). The addition of the linear interpretable model before the TCN slightly improves the test error. We note this effect is more visible on A2, A3, A4, mainly due to the non-stationary nature of these datasets and the fact that TCNs do not easily approximate trends (Braei & Wagner, 2020) (we further tested this

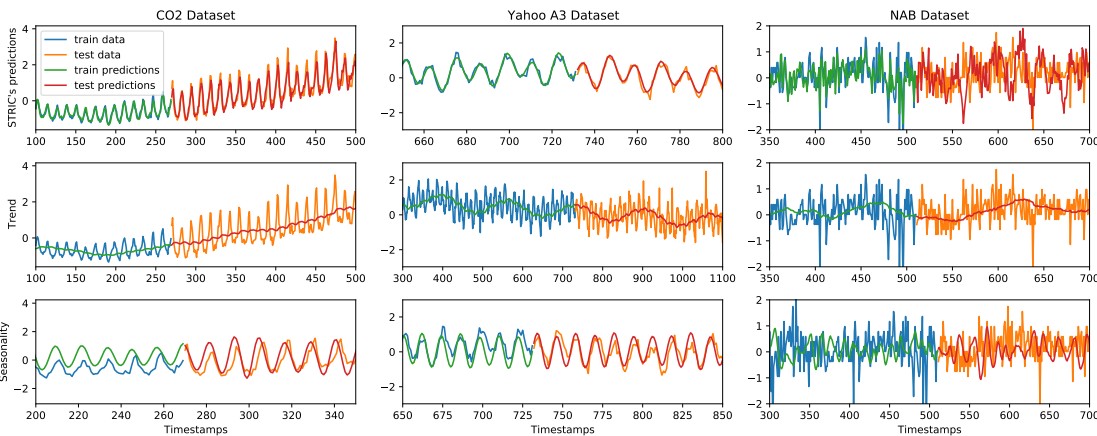

Figure 2: We test STRIC time series intepretability on different datasets (columns). In each panel, we show both training data and test data (see colors). **First row**: STRIC time series predictor (output of non-linear module). **Second row**: Trend components extracted by the interpretable blocks. **Third row**: Seasonal components extracted by the interpretable blocks.

Table 2: **Ablation study on the RMSE of prediciton errors**: We compare Test error and Generalization Gap (Gap.) of a standard TCN model with our STRIC predictor and some variation of it (using the same training hyper-parameters). Standard deviations are given in Table 4.

|  |  | TCN | | TCN + Linear | | TCN + Fading | | STRIC pred | |
|---|---|---|---|---|---|---|---|---|---|
|  |  | Test | Gap. | Test | Gap. | Test | Gap. | Test | Gap. |
| **Datasets** | Yahoo A1 | 0.92 | 0.82 | 0.88 | 0.78 | 0.92 | 0.48 | **0.62** | **0.19** |
|  | Yahoo A2 | 0.82 | 0.71 | 0.35 | 0.22 | 0.71 | 0.50 | **0.30** | **0.16** |
|  | Yahoo A3 | 0.43 | 0.30 | **0.22** | 0.06 | 0.40 | 0.25 | **0.22** | **0.03** |
|  | Yahoo A4 | 0.61 | 0.46 | 0.35 | 0.16 | 0.55 | 0.38 | **0.24** | **0.01** |
|  | CO2 Dataset | 0.62 | 0.48 | 0.45 | 0.30 | 0.61 | 0.43 | **0.41** | **0.08** |
|  | NAB Traffic | 1.06 | 1.03 | 1.00 | 0.96 | 0.93 | 0.31 | **0.74** | **0.11** |
|  | NAB Tweets | 1.02 | 0.84 | 0.98 | 0.78 | 0.83 | 0.36 | **0.77** | **0.07** |

in Appendix A.7.1). While STRIC generalization is always better than a standard TCN model and STRIC's ablated components, we note that applying Fading memory regularization alone to a standard TCN does not always improve generalization (but never decreases it): this highlights that the benefits of combining the linear module and the fading regularization together are not a trivial 'sum of the parts'. Consider for example Yahoo A1: STRIC achieves 0.62 test error, the best ablated model (TCN + Linear) 0.88, while TCN + Fading does not improve over the baseline TCN. A similar observation holds for the CO2 Dataset. Fading regularization might not be beneficial (nor detrimental) for time series containing purely periodic components which correspond to infinite memory systems (systems with unitary fading coefficient). In such cases the interpretable module is essential in removing the periodicities and providing the regularized non-linear module (TCN + Fading) with an easier to model residual signal. We refer to Figure 2 (first column) for a closer look on a typical time series in CO2 dataset, which contains a periodic component that is captured by the seasonal part of the interpretable model. To conclude, our proposed fading regularization has (on average) a beneficial effect in controlling the complexity of a standard TCN model and reduces its generalization gap ($\approx 40\%$ reduction). Moreover, coupling fading regularization with the interpretable module guarantees the best generalization.

**Automatic complexity selection**: In Figure 3, we test the effects of our automatic complexity selection (fading memory regularization) on STRIC. We compare STRIC with a standard TCN model and STRIC without regularization as the memory of the predictor increases. The test error of STRIC is uniformly smaller than a standard TCN (without interpretable blocks nor fading regularization). Adding interpretable blocks to a standard TCN improves generalization for a fixed memory w.r.t. standard TCN but gets worse (overfitting occurs) as soon as the available past data horizon increases. On the other hand, the generalization gap of STRIC does not deteriorate as the memory of the predictor increases (see Appendix A.7.1 for a comparison with other metrics).

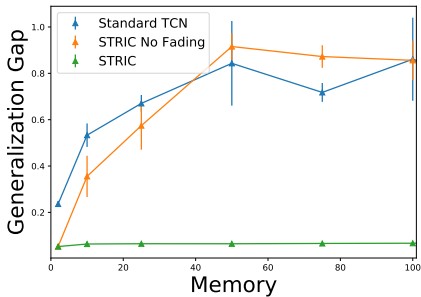

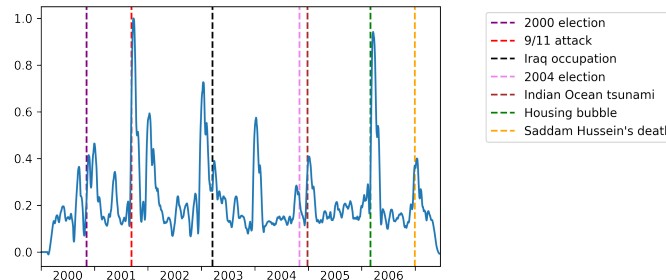

Figure 3: **Automatic complexity selection**: Fading memory regularization preserves generalization gap as the memory of the predictor $n_p$ increases on NAB Tweets.

Figure 4: **Anomaly score on the New York Times dataset.** Our method finds anomalies in a complex time series consisting of the BERT embedding of articles from the New York Times. Peaks in the anomaly score correspond to historical events that sensibly changed the content of the news cycle.

**Anomaly detection on the New York Times dataset**: We qualitatevly test STRIC on a time series consisting of BERT embeddings (Devlin et al., 2019) of New York Times articles (Sandhaus, 2008) from 2000 to 2007. We set $n_p = n_f = 30$ days, to be able to detect change-point anomalies that altered the normal distribution of news articles for a prolonged period of time. Without any human annotation, STRIC is able to detect major historical events such as the 9/11 attack, the 2004 Indian Ocean tsunami, and U.S. elections (Figure 4). Note that we do not carry out a quantitative analysis of STRIC's predictions, as we are not aware of any ground truth or metrics for this benchmark, see for example Rayana & Akoglu (2015). Additional details and comparison with a baseline model built on PCA are given in Appendix A.8.1.

## 7 DISCUSSION AND CONCLUSIONS

We have shown that our interpretable stacked residual architecture and our unsupervised estimation of the likelihood ratio are well suited to solve AD for multivariate time series data. Unlike purely DNN-based methods (Geiger et al., 2020; Munir et al., 2019; Bashar & Nayak, 2020), STRIC exposes to the user both an interpretable STL-like time series decomposition (Cleveland et al., 1990) and the relevant time scale of the time series. Both the interepretable module and the fading memory regularization are important in building a successful model (see Table 2). In particular, the interpretable module helps STRIC generalize correctly on non-stationary time series on which standard deep models (such as TCNs) may overfit (Braei & Wagner, 2020). Moreover, we showed that our novel fading regularization alone can improve the generalization error of TCNs up to $\approx 40\%$ over standard TCNs, provided the periodic components of the time series are captured by the interpretable module (Section 6). We highlight that the overall computational complexity and memory requirement of our method remains the same as standard TCNs, so that our approach can easily scale to large scale time series datasets.

An anomaly is a *time instant*: at that time instant, either we receive an isolated observation that is inconsistent with normal operation (outlier measurement), or a discrete change occurs in the mechanism that generates the data (change-point) that persists beyond that time instant (Geiger et al., 2020; Basseville & Nikiforov, 1993). Our method treats these two phenomena in a unified manner, without the need to differentiate between outliers and setpoint changes, with specialized detectors for each. Once the predictor is built, our method can be used online to detect anomalies soon after occurrence without waiting for the entire data stream to be observed and without requiring any knowledge on the prediction error distribution (nominal and faulty).

Making the non-parametric anomaly detector fully adaptive to the data is an interesting research direction: While our fading window regularizer automatically tunes the predictor's window length by exploiting the self-supervised nature of the prediction task, methods to automatically tune the detector's window lengths (an unsupervised problem) are an interesting research direction. Moreover, designing statistically optimal rules to calibrate our detector's threshold $\tau$ depending on the desired operating point in the tradeoff between missed detection and false alarms would further enhance the out-of-the-box robustness of our method.

**Reproducibility Statement**: Our method has been tested on publicly available datasets: Yahoo (Laptev & Amizadeh, 2020), NAB (Lavin & Ahmad, 2015), CO2 Dataset (see Appendix A.6) and NYT (Sandhaus, 2008). We described the data splitting and the data processing steps in Appendix A.7. We describe the major details regarding the implementation of our novel method in Appendix A.1 while in Appendix A.7 we describe both the model structure and training hyper-parameters we used in the experimental section. We do not include the model structures and hyper-parameters of SOTA methods we used as baselines and refer to related literature (referenced in our work) for the implementation details. To further foster reproducibility we shall make our code available.

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

# A  APPENDIX

## A.1  IMPLEMENTATION

Given two scalar time series $x$ and $\varphi$, we denote their time convolution $g$ as: $g(t) := (\varphi * x)(t) := \sum_{i=-\infty}^{\infty} \varphi(i)x(t-i)$. We say that $g$ is the causal convolution of $x$ and $\varphi$ if $\varphi(t) = 0$ for $t < 0$, so that the output $g(t)$ does not depend on future values of $x$ (w.r.t. current time index $t$). In the following, we shall give a particular interpretation to the signals $x$ and $\varphi$: $x$ will be the input signal to a filter parametrized by an impulse response $\varphi$ (kernel of the filter). Note any (causal) convolution is defined by an infinite summation for each given time $t$. Therefore it is customary, when implementing convolutional filters, to consider a truncated sum of finite length. In practice, this is obtained assuming the filter impulse response is non-zero only in a finite window. The truncation is indeed an approximation of the complete convolution, nonetheless it is possible to prove that the approximation errors incurred due to truncation are guaranteed to be bounded under simple assumptions. To summarize, in the following we shall write $g(t) := (\varphi * x)(t)$ and mean that the impulse response of the causal filter $\varphi$ is truncated on a window of a given length.

### A.1.1  ARCHITECTURE

Let $Y_{t-n_p+1}^t \in \mathbb{R}^{n \times n_p}$ be the input data to our architecture at time step $t$ (a window of $n_p$ past time instants). The main blocks of the architecture are defined to encode trend, seasonality, stationary linear and non-linear part. In the following we shall denote each quantity related to a specific layer using either the subscripts {TREND, SEAS, LIN, TCN} or {0, 1, 2, 3}.

We shall denote the input of each block as $X_k \in \mathbb{R}^{n \times n_p}$ and the output as $\hat{X}_k \in \mathbb{R}^{n \times n_p}$ for $k = 0, 1, 2, 3$. The residual architecture we propose is defined by the following: $X_0 = Y_{t-n_p+1}^t$ and $X_k = X_{k-1} - \hat{X}_{k-1}$ for $k = 1, 2, 3$. At each layer we extract $l_k$ temporal features from the input $X_k$. We denote the temporal features extracted from the input of the $k$-th block as: $G_k := G_k(X_k) \in \mathbb{R}^{l_k \times n_p}$. The $i$-th column of the feature matrix $G_k$ is a feature vector (of size $l_k$) extracted from the input $X_k$ up to time $t - n_p - i$. To do so, we use causal convolutions of the input signal $X_k$ with a set of filter banks (Bai et al., 2018).

**Modeling Interpretable blocks**: In this section, we shall describe the main design criteria of the linear module. For each interpretable layer (TREND, SEAS, LIN), we convolve the input signal with a filter bank designed to extract specific components of the input.

For example, consider the trend layer, denoting its scalar input time series by $x$ and its output by $g_{\text{TREND}}$. Then $g_{\text{TREND}}$ is defined as a multidimensional time series (of dimension $l_{\text{TREND}} := l_0$) obtained by stacking $l_0$ time series given by the convolution of $x$ with $l_0$ causal linear filters: $\varphi_{\text{TREND}i} * x$ for $i = 0, ..., l_0 - 1$. In other words, $g_{\text{TREND}} := [\varphi_{\text{TREND}1} * x, \varphi_{\text{TREND}2} * x, ..., \varphi_{\text{TREND}l_1} * x]^T$. We denote the set of linear filters $\varphi_{\text{TREND}i}$ for $i = 0, ..., l_0 - 1$ as $\mathcal{K}_{\text{TREND}}$ and parametrize each filter in $\mathcal{K}_{\text{TREND}}$ with its truncated impulse response (i.e. kernel) of length $k_0 := k_{\text{TREND}}$.

We interpret each time series in $g_{\text{TREND}}$ as an approximation of the trend component of $x$ computed with the $i$-th filter. We design each $\varphi_{\text{TREND}i}$ so that each filter extracts the trend of the input signal on different time scales (Ravn & Uhlig, 2002) (i.e., each filter outputs a signal with a different smoothness degree). We estimate the trend of the input signal by recombining the extracted trend components in $g_{\text{TREND}}$ with the linear map $a_{\text{TREND}}$. Moreover, we predict the future trend of the input signal (on the next time-stamp) with the linear map $b_{\text{TREND}}$.

We construct the blocks that extract seasonality and linear part in a similar way.

**Implementing Interpretable blocks**: The input of each layer is given by a window of measurements of length $n_p$. We zero-pad the input signal so that the convolution of the input signal with the $i$-th filter is a signal of length $n_p$ (note this introduces a spurious transient whose length is the length of the filter kernel). We therefore have the following temporal feature matrices: $G_0 = G_{\text{TREND}} \in \mathbb{R}^{l_0 \times n_p}$, $G_1 = G_{\text{SEAS}} \in \mathbb{R}^{l_1 \times n_p}$ and $G_2 = G_{\text{LIN}} \in \mathbb{R}^{l_2 \times n_p}$.

The output of each layer $\hat{X}_k$ is an estimate of the trend, seasonal or stationary linear component of the input signal on the past interval of length $n_p$, so that we have $\hat{X}_k \in \mathbb{R}^{1 \times n_p}$ (same dimension as the input $X_k$). On the other hand, the linear predictor $\hat{y}_k$ computed at each layer is a scalar. Intuitively, $\hat{X}_k$ and $\hat{y}_k$ should be considered as the best linear approximation of the trend, seasonality or linear part given block's filter bank in the past and future. Our architecture performs the following computations: $\hat{X}_k := a_k^T G_k$ and $\hat{y}_k := \hat{X}_k b_k$ for $k = 0, 1, 2$ where $a_i \in \mathbb{R}^{l_k}$ and $b_k \in \mathbb{R}^{n_p}$. Note $a_k$ combines features (uniformly in time) so that we can interpret it as a feature selector while $b_k$ aggregates relevant features across different time indices to build the one-step ahead predictor. Depending on the time scale of the signals it is possible to choose $b_k$ depending on the time index (similarly to the fading memory regularization). We experimented both the choice to make $b_k$ canonical "vectors" and dense vectors. We found that choosing $b_k$ as canonical vectors, whose non-zero entry is associated to the closest to the present time instat provides good empirical results on most cases.

**Non-linear module** The non-linear module is based on a standard TCN network. Its input is defined as $X_3 = Y_{t-n_p+1}^t - \hat{X}_0 - \hat{X}_1 - \hat{X}_2$, which is to be considered as a signal whose linearly predictable component has been removed. The TCN extracts a set of $l_3$ non-linear features $G_3(X_3) \in \mathbb{R}^{l_3 \times n_p}$ which we combine with linear maps as done for the previous layers. The $j$-th column of the non-linear features $G_3$ is computed using data up to time $t - n_p + j$ (due to the internal structure of a TCN network (Bai et al., 2018)). The linear predictor on top of $G_3$ is $\hat{y}_{\text{TCN}} := a_3^T G_3 b_3$, where $a_3 \in \mathbb{R}^{l_3}$ and $b_3 \in \mathbb{R}^{n_p}$.

Finally, the output of our time model is given by:

$$\hat{y}(t+1) = \sum_{k=0}^{3} \hat{y}_k = \sum_{k=0}^{3} \hat{X}_k b_k = \sum_{k=0}^{3} a_k^T G_k(X_k) b_k.$$

We extend our architecture to multi-dimensional time series according to the following principles: preserve interpretability (first module) and exploit global information to make local predictions (second module).

In this section, the input data to our model is $Y_{t-n_p+1}^t \in \mathbb{R}^{n \times n_p}$ (a window of length $n_p$ from an $n$-dimensional time series).

**Interpretable module**: Each time series undergoes the sequence of 3 interpretable blocks independently from other time series: the filter banks are applied to each time series independently. Therefore, each time series is processed by

the same filter banks: $\mathcal{K}_{\text{TREND}}$, $\mathcal{K}_{\text{SEAS}}$ and $\mathcal{K}_{\text{LIN}}$. For ease of notation we shall now focus only on the trend layer. Any other layer is obtained by substituting 'TREND' with the proper subscript ('SEAS' or 'LIN').

We denote by $G_{\text{TREND}}, i \in \mathbb{R}^{l_0 \times n_p}$ the set of time features extracted by the trend filter bank $\mathcal{K}_{\text{TREND}}$ from the $i$-th time series. Each feature matrix is then combined as done in the scalar setting using linear maps, which we now index by the time series index $i$: $a_{\text{TREND}i}$ and $b_{\text{TREND}i}$. The rationale behind this choice is that each time series can exploit differently the extracted features. For instance, slow time series might need a different filter than faster ones (chosen using $a_{\text{TREND}i}$) or might need to look at values further in the past (retrieved using $b_{\text{TREND}i}$). We stack the combination vectors $a_{\text{TREND}i}$ and $b_{\text{TREND}i}$ into the following matrices: $\mathcal{A}_{\text{TREND}} = [a_{\text{TREND}1}, a_{\text{TREND}2}, ..., a_{\text{TREND}n}]^T \in \mathbb{R}^{n \times l_0}$ and $\mathcal{B}_{\text{TREND}} = [b_{\text{TREND}1}, b_{\text{TREND}2}, ..., b_{\text{TREND}n}]^T \in \mathbb{R}^{n \times n_p}$.

**Non-linear module**: The second (*non-linear*) module aggregates global statistics from different time series (Sen et al., 2019) using a TCN model. It takes as input the prediction residual of the linear module and outputs a matrix $G_{\text{TREND}}(Y_{t-n_p+1}^t) \in \mathbb{R}^{l_3 \times n_p}$ where $l_3$ is the number of output features that are extracted by the TCN model (which is a design parameter). The $j$-th column of the non-linear features $G_{\text{TREND}}(Y_{t-n_p+1}^t)$ is computed using data up to time $t - p + j$, where $p$ is the "receptive" field of the TCN ($p < n_p$). This is due to the internal structure of a TCN network (Bai et al., 2018) which relies on causal convolutions and typically scales as $O(2^h)$ where $h$ is the number of TCN hidden layers (the deeper the TCN the longer its receptive field). As done for the time features extracted by the interpretable blocks, we build a linear predictor on top of $G_{\text{TREND}}(Y_{t-n_p+1}^t)$ for each single time series independently: the predictor for the $i$-th time series is given by: $\hat{y}_{\text{TCN}}(t+1)_i := a_i^T G_{\text{TREND}}(Y_{t-n_p+1}^t) b_i$ where $a_i \in \mathbb{R}^{l_3}$ and $b_i \in \mathbb{R}^{n_p}$. We stack the combination vectors $a_{\text{TCN}i}$ and $b_{\text{TCN}i}$ into the following matrices: $\mathcal{A}_{\text{TCN}} = [a_{\text{TCN}1}, a_{\text{TCN}2}, ..., a_{\text{TCN}n}]^T \in \mathbb{R}^{n \times l_3}$ and $\mathcal{B}_{\text{TCN}} = [b_{\text{TCN}1}, b_{\text{TCN}2}, ..., b_{\text{TCN}n}]^T \in \mathbb{R}^{n \times n_p}$.

Finally, the outputs of the predictor on the $i$-th time series are given by:

$$\hat{y}(t+1)_i = \sum_{k \in \{\text{TREND,SEAS,LIN}\}} a_{ki}^T G_{ki} b_{ki} + a_{\text{TCN}i}^T G_{\text{TCN}} b_{\text{TCN}i}.$$

### A.1.4 BLOCK STRUCTURE AND INITIALIZATION

In this section, we shall describe the internal structure and the initialization of each block.

**Structure**: Each filter is implemented by means of depth-wise causal 1-D convolutions (Bai et al., 2018). We call the tensor containing the $k$-th block's kernel parameters $\mathcal{K}_k \in \mathbb{R}^{l_k \times N_k}$, where $l_k$ and $N_k$ are the block's number of filters and block's kernel size, respectively (without loss of generality, we assume all filters have the same dimension). Each filter (causal 1D-convolution) is parametrized by the values of its impulse response parameters (kernel parameters). When we learn a filter bank, we mean that we optimize over the kernel values for each filter jointly. For multidimensional time series, we apply the filter banks to each time series independently (depth-wise convolution) and improve filter learning by sharing kernel parameters across different time series.

**Initialization**: The *first block* (trend) is initialized using $l_0$ causal Hodrick Prescott (HP) filters (Ravn & Uhlig, 2002) of kernel size $N_0$. HP filters are widely used to extract trend components of signals (Ravn & Uhlig, 2002). In general a HP filter is used to obtain from a time series a smoothed curve which is not sensitive to short-term fluctuations and more sensitive to long-term ones (Ravn & Uhlig, 2002). In general, a HP filter is parametrized by a hyper-parameter $\lambda_{\text{HP}}$ which defines the regularity of the filtered signal (the higher $\lambda_{\text{HP}}$, the smoother the output signal). We initialize each filter with $\lambda_{\text{HP}}$ chosen uniformly in log-scale between $10^3$ and $10^9$. Note the impulse response of these filters decays to zero (i.e., the latest samples from the input time series are the most influential ones). When we learn the optimal set of trend filter banks, we do not consider them parametrized by $\lambda_{\text{HP}}$ and search for the optimal $\lambda_{\text{HP}}$. Instead, we optimize over the impulse response parameters of the kernel which we do not assume live in any manifold (e.g., the manifold of HP filters). Since this might lead to optimal filters which are not in the class of HP filters, we impose a regularization which penalizes the distance of the optimal impulse response parameters from their initialization.

The *second block* (seasonal part) is initialized using $l_1$ periodic kernels which are obtained as linear filters whose poles (i.e., frequencies) are randomly chosen on the unit circle (this guarantees to span a range of different frequencies). Note the impulse responses of these filters do not go to zero (their memory does not fade away). Similarly to the HP filter bank, we do no optimize the filters over frequencies, but rather we optimize them over their impulse response (kernel parameters). This optimization does not preserve the strict periodicity of filters. Therefore, in order to keep the optimal impulse response close to initialization values (purely periodic), we exploit weight regularization by penalizing the distance of the optimal set of kernel values from initialization values.

The *third block* (stationary linear part) is initialized using $l_2$ randomly chosen linear filters whose poles lie inside the unit circle, as done in (Farahmand et al., 2017). As the number of filters $l_2$ increases, this random filter bank is guaranteed to be a universal approximator of any (stationary) linear system (see (Farahmand et al., 2017) for details).

**Remark**: This block could approximate any trend and periodic component. However, we assume to have factored out both trend and periodicities in the previous blocks.

The last module (*non-linear part*) is composed by a randomly initialized TCN model. We employ a TCN model due to its flexibility and capability to model both long-term and short-term non-linear dependencies. As is standard practice, we exploit dilated convolutions to increase the receptive field and make the predictions of the TCN (on the future horizon) depend on the most relevant past (Bai et al., 2018).

**Remark**: Our architecture provides an interpretable justification of the initialization scheme proposed for TCN in (Sen et al., 2019). In particular our convolutional architecture allows us to handle high-dimensional time series data without a-priori standardization (e.g., trend or seasonality removal).

### A.2   AUTOMATIC COMPLEXITY DETERMINATION (FADING MEMORY REGULARIZATION)

In this section, we shall introduce a regularization scheme called *fading regularization*, to constrain TCN representational capabilities.

The output of the TCN model is $G(Y_{t-n_p+1}^t) \in \mathbb{R}^{l_3 \times n_p}$ where $l_3$ is the number of output features extracted by the TCN model. The predictor build from TCN features is given by: $a_{\text{TCN}i}^T G_{\text{TCN}}(Y_{t-n_p+1}^t) b_{\text{TCN}i}$, where the predictor $b_{\text{TCN}i} \in \mathbb{R}^{n_p}$ takes as input a linear combination of the TCN features (weighted by $a_{\text{TCN}i}$). The $j$-th column of the non-linear features $G(Y_{t-n_p+1}^t)$ is computed using data up to time $t - n_p + j$ (due to causal convolutions used in the internal structure of the TCN network (Bai et al., 2018)). One expects that the influence on the TCN predictor as $j$ increases should increase too (in case $j = n_p$, the statistic is the one computed on the closest window of time w.r.t. present time stamp). Clearly, the exact *relevance* on the output is not known a priori and needs to be estimated. In other words, the predictor should be less sensitive to statistics (features) computed on a far past, a property which is commonly known as *fading memory*. Currently, this property is not built in the predictor $b_{\text{TCN}i}$, which treats each time instant equally and might overfit while trying to explain the future by looking into far and possibly non-relevant past. In order to constrain model complexity and reduce overfitting, we impose the fading memory property on our predictor by employing a specific regularization which we now describe.

### A.2.1   FADING MEMORY IN SCALAR TIME SERIES

We now follow the same notation and assumptions used in Section 4.1 which we now repeat for completeness.

We consider a scalar time series so that the TCN-based future predictor given the past $n_p$ measures can be written as: $\hat{y}_{\text{TCN}}(t+1) = a^T G_{\text{TCN}}(T_{t-n_p+1}^t) b = \hat{X}_k b$. We shall assume that innovations (optimal prediction errors) are Gaussian, so that $y(t+1) \mid Y_{t-n_p+1}^t \sim \mathcal{N}(F^*(Y_{t-n_p+1}^t)), \eta^2)$, where $F^*$ is the optimal predictor of the future values given the past. Note that this assumption does not restrict our framework and is used only to justify the use of the squared loss to learn the regression function of the predictor. In practice, we do not know the optimal $F^*$ and we approximate it with our parametric model. For ease of exposition, we group all the architecture parameters except $b$ in the weight vector $W$ (linear filters parameters $\mathcal{K}_{\text{TREND}}, \mathcal{K}_{\text{SEAS}}, \mathcal{K}_{\text{LIN}}$, linear module recombination weights $\mathcal{A}_{\text{TREND}}, \mathcal{A}_{\text{SEAS}}, \mathcal{A}_{\text{LIN}}, \mathcal{B}_{\text{TREND}}, \mathcal{B}_{\text{SEAS}}, \mathcal{B}_{\text{LIN}}$, and TCN kernel parameters and recombination coefficients $\mathcal{A}_{\text{TCN}}$ etc.). We write the conditional likelihood of the future given the past data of our parametric model as:

$$p(Y_{t+1}^{t+n_f} \mid b, W, Y_{t-n_p+1}^t) = \prod_{k=1}^{n_f} p(y(t+k) \mid b, W, Y_{t+k-n_p}^{t+k-1}) \tag{6}$$

To make the notation simpler, we shall denote by $Y_f := Y_{t+1}^{t+n_f} \in \mathbb{R}^{n_f}$ the set of future outputs over which the predictor is computed and we shall use $\hat{Y}_{b,W} \in \mathbb{R}^{n_f}$ as the predictor's outputs. Moreover, we shall drop the dependency on the conditioning past $Y_{t-n_p+1}^t$ (which is present in any conditional distribution). Equation (6) becomes: $p(Y_f \mid b, W) = \prod_{k=1}^{n_f} p(y(t+k) \mid b, W)$. The optimal set of parameters $b^*$ and $W^*$ in a Bayesian framework is computed by maximizing the posterior on the parameters given the data:

$$p(b, W \mid Y_f) \propto p(Y_f \mid b, W) p(b) p(W) \tag{7}$$

where $p(b)$ is the prior on the predictor and $p(W)$ is the prior on the remaining parameters. We encode in $p(b)$ our prior belief that the complexity of the predictor should not be too high and therefore it should only depend on the *most relevant past*.

**Remark**: The prior does not induce hard constraints. It rather biases the optimal predictor coefficients towards the prior belief. This is clear by looking at the negative log-posterior which can be directly interpreted as the loss function to be minimized: $-\log p(b, W \mid Y_f) = -\log p(Y_f \mid b, W) - \log p(b) - \log p(W)$. In particular, the first term $\log p(Y_f \mid b, W)$ is the data fitting term (only influenced by the data). Both $\log p(b)$ and $\log p(W)$ do not depend on the available data and can be interpreted as regularization terms that bound the complexity of the predictor function.

The main idea is to reduce the sensitivity of the predictor on time instants that are far in the past. We therefore enforce the *fading memory* assumption on $p(b)$ by assuming that the components of $b \in \mathbb{R}^{n_p}$ have zero mean and exponentially decaying variances:

$$\mathbb{E}b_j = 0 \text{ and } \mathbb{E}b_{n_p-j-1}^2 = \kappa\lambda^j \text{ for } j = 0, ..., n_p - 1 \tag{8}$$

where $\kappa \in \mathbb{R}^+$ and $\lambda \in (0, 1)$. Note the larger variance (larger scale) is associated to temporal indices close to the present time $t$.

**Remark**: To specify the prior, we need a density function $p(b)$ but up to now we only specified constraints on the first and second order moments. We therefore need to constrain the parametric family of prior distributions we consider. Any choice on the class of prior distributions lead to different optimal estimators. Among all the possible choices of prior families we choose the maximum entropy prior (Cover & Thomas, 1991). Under constraints on first and second moment, the maximum entropy family of priors is the exponential family (Cover & Thomas, 1991). In our setting, we can write it as:

$$\log p_{\lambda,\kappa}(b) \propto -\|b\|_{\Lambda^{-1}}^2 - \log|\Lambda| \tag{9}$$

where $\Lambda \in \mathbb{R}^{n_p \times n_p}$ is a diagonal matrix whose elements are $\Lambda_{j,j} = \kappa\lambda^j$ for $j = 0, ..., n_p - 1$.

The parameter $\lambda$ represents how fast the predictor's output 'forgets' the past: the smaller $\lambda$, the lower the complexity. In practice, we do not have access to this information and indeed we need to estimate $\lambda$ from the data.

One would be tempted to estimate jointly $W, b, \lambda, \kappa$ (and possibly $\eta$) by minimizing the negative log of the joint posterior:

$$\arg\min_{b,W,\lambda,\kappa} \frac{1}{\eta^2} \left\| Y_f - \hat{Y}_{b,W} \right\|^2 + \log(\eta^2) - \log(p_{\lambda,\kappa}(B)) - \log(p(W)). \tag{10}$$

Unfortunately, this leads to a degeneracy since the joint negative log posterior goes to $-\infty$ when $\lambda \to 0$.

**Bayesian learning formulation for fading memory regularization**:

The parameters describing the prior (such as $\lambda$) are typically estimated by maximizing the marginal likelihood, i.e., the likelihood of the data once the parameters $(b, W)$ have been integrated out. Unfortunately, the task of computing (or even approximating) the marginal likelihood in this setup is prohibitive and one would need to resort to Monte Carlo sampling techniques. While this is an avenue worth investigating, we preferred to adopt the following variational strategy inspired by the linear setup.

Indeed, the model structure we consider is linear in $b$ and we can therefore stack the predictions of each available time index $t$ to get the following linear predictor on the whole future data: $\hat{Y}_{b,W} = F_W b$ where $F_W \in \mathbb{R}^{n_f \times n_p}$ and its rows are given by $\hat{X}_{\text{TCN}}(Y_{i-n_p+1}^i)$ for $i = t, ..., t + n_f - 1$.

We are now ready to find an upper bound to the marginal likelihood associated to the posterior given by Equation (7) with marginalization taken only w.r.t. $b$.

**Proposition A.1** (from (Tipping, 2001)). *The optimal value of a regularized linear least squares problem with feature matrix $F$ and parameters $b$ is given by the following equation:*

$$\arg\min_b \frac{1}{\eta^2}\|Y_f - Fb\|^2 + b^\top\Lambda^{-1}b = Y_f^\top\Sigma^{-1}Y_f \tag{11}$$

*with $\Sigma := F\Lambda F^\top A^T + \eta^2 I$.*

Equation (11) guarantees that

$$\frac{1}{\eta^2}\|Y_f - Fb\|^2 + b^\top\Lambda^{-1}b + \log|\Sigma| \geq Y_f^\top\Sigma^{-1}Y_f + \log|\Sigma|,$$

where the right hand side is (proportional to) the negative marginal likelihood with marginalization taken *only* w.r.t. $b$. Therefore, for fixed a $W$,

$$\frac{1}{\eta^2} \left\| Y_f - \hat{Y}_{b,W} \right\|^2 + b^\top \Lambda^{-1} b + \log |F_W \Lambda F_W^\top + \eta^2 I|$$

is an upper bound of the marginal likelihood with marginalization over $b$ and does not suffer of the degeneracy alluded at before.

With this considerations in mind, and inserting back the optimization over $W$, the overall optimization problem we solve is

$$\underset{b,W,\lambda \in (0,1),\kappa > 0}{\arg \min} \frac{1}{\eta^2} \left\| Y_f - \hat{Y}_{b,W} \right\|^2 + \|b\|_{\Lambda^{-1}}^2 + \log |F_W \Lambda F_W + \eta^2 I| + \log p(W) \tag{12}$$

**Remark**: $\log p(W)$ defines the regularization applied on the remaining parameters of our architecture. In particular, we induce sparsity by applying $L^1$ regularization on $\mathcal{A}_{\text{TREND}}$, $\mathcal{A}_{\text{SEAS}}$, $\mathcal{A}_{\text{LIN}}$ and $\mathcal{A}_{\text{TCN}}$. Also, we constrain filters parameters to stay close to initialization by applying $L^2$ regularization on $\mathcal{K}_{\text{TREND}}$, $\mathcal{K}_{\text{SEAS}}$ and $\mathcal{K}_{\text{LIN}}$.

### A.2.2 FADING MEMORY IN MULTIVARIATE TIME SERIES

In the case of multivariate time series, fading regularization can be applied either with a single fading coefficient $\lambda$ for all the time series or with different fading coefficients for each time series. In all the experiments in this paper, we chose to keep one single $\lambda$ for all the time series. In practice, this choice is sub-optimal and might lead to more overfitting than treating each time series separately: the 'dominant' (slower) time series will highly influence the optimal $\lambda$.

### A.2.3 FEATURES NORMALIZATION

We avoid the non-identifiability of the product $F_W b$ by exploiting batch normalization: we impose that different features have comparable means and scales across time indices $i = 0, ..., n_p - 1$. Non-identifiability occurs due to the product $F_W b$, if features have different scales across time indices (i.e., columns of the matrix $F_W$) the benefit of fading regularization might reduced since it can happen that features associated with small $b_i$ have large scale so that the overall contribution of the past does not fade. Hence we use batch normalization to normalize time features. Then we use an affine transformation (with parameters to be optimized) to jointly re-scale all the output blocks before the linear combination with $b$.

### A.3 ALTERNATIVE CUMSUM DERIVATION AND INTERPRETATION

In this section, we describe an equivalent formulation of the CUMSUM algorithm we derived in the main paper. Before a change point, by construction we are under the distribution of the past. Therefore, $\log \frac{p_f(y)}{p_p(y)} \leq 0 \ \forall y$, which in turn means that the cumulative sum $S_1^t$ will decrease as $t$ increases (negative drift). After the change, the situation is opposite and the cumulative sum starts to show a positive drift, since we are sampling $y(i)$ from the future distribution $p_f$. This intuitive behaviour shows that the relevant information to detect a change point can be obtained directly from the cumulative sum (along timestamps). In particular, all we need to know is the difference between the value of the cumulative sum of log-likelihood ratios and its minimum value.

The CUMSUM algorithm can be expressed using the following equations: $v_t := S_1^t - m_t$, where $m_t := \min_{j,1 \leq j \leq t} S_j^t$. The stopping time is defined as: $t_{\text{stop}} = \min\{t : v_t \geq \tau\} = \min\{t : S_1^t \geq m_t + \tau\}$. With the last equation, it becomes clear that the CUMSUM detection equation is simply a comparison of the cumulative sum of the log likelihood ratios along time with an adaptive threshold $m_t + \tau$. Note that the adaptive threshold keeps complete memory of the past ratios. The two formulations are equivalent because $S_1^t - m_t = h_t$.

### A.4 VARIATIONAL APPROXIMATION OF THE LIKELIHOOD RATIO

In this section, we present some well known facts on $f$-divergences and their variational characterization. Most of the material and the notation is from (Nguyen et al., 2010). Given a probability distribution $\mathbb{P}$ and a random variable $f$ measurable w.r.t. $\mathbb{P}$, we use $\int f d\mathbb{P}$ to denote the expectation of $f$ under $\mathbb{P}$. Given samples $x(1), ..., x(n)$ from $\mathbb{P}$, the

empirical distribution $\mathbb{P}_n$ is given by $\mathbb{P}_n = \frac{1}{n} \sum_{i=1}^{n} \delta_{x(i)}$. We use $\int f d\mathbb{P}_n$ as a convenient shorthand for the empirical expectation $\frac{1}{n} \sum_{i=1}^{n} f(x(i))$.

Consider two probability distributions $\mathbb{P}$ and $\mathbb{Q}$, with $\mathbb{P}$ absolutely continuous w.r.t. $\mathbb{Q}$. Assume moreover that both distributions are absolutely continuous with respect to the Lebesgue measure $\mu$, with densities $p_0$ and $q_0$, respectively, on some compact domain $\mathcal{X} \subset \mathbb{R}^d$.

**Variational approximation of the f-divergence**: The $f$-divergence between $\mathbb{P}$ and $\mathbb{Q}$ is defined as (Nguyen et al., 2010)

$$D_f(\mathbb{P}, \mathbb{Q}) := \int p_0 f\left(\frac{q_0}{p_0}\right) d\mu \tag{13}$$

where $f : \mathbb{R} \to \mathbb{R}$ is a convex and lower semi-continuous function. Different choices of $f$ result in a variety of divergences that play important roles in various fields (Nguyen et al., 2010). Equation (13) is usually replaced by the variational lower bound:

$$D_f(\mathbb{P}, \mathbb{Q}) \geq \sup_{\phi \in \Phi} \int [\phi d\mathbb{Q} - f^*(\phi) d\mathbb{P}] \tag{14}$$

and equality holds iff the subdifferential $\partial f\left(\frac{q_0}{p_0}\right)$ contains an element of $\Phi$. Here $f^*$ is defined as the convex dual function of $f$.

In the following, we are interested in divergences whose conjugate dual function is smooth (which in turn defines commonly used divergence measures such as KL and Pearson divergence), so that we shall assume that $f$ is convex and differentiable. Under this assumption, the notion of subdifferential is not required and the previous statement reads as: *equality holds iff $\partial f\left(\frac{q_0}{p_0}\right) = \phi$ for some $\phi \in \Phi$.*

**Remark**: The infinite-dimensional optimization problem in Equation (14) can be written as $D_f(\mathbb{P}, \mathbb{Q}) = \sup_{\phi \in \Phi} \mathbb{E}_{\mathbb{Q}} \phi - \mathbb{E}_{\mathbb{P}} f^*(\phi)$.

In practice, one can have an estimator of any $f$-divergence restricted to a functional class $\Phi$ by solving Equation (14) (Nguyen et al., 2010). Moreover, when $\mathbb{P}$ and $\mathbb{Q}$ are not known one can approximate them using their empirical counterparts: $\mathbb{P}_n$ and $\mathbb{Q}_n$. Then an empirical estimate of the $f$-divergence is: $\hat{D}_f(\mathbb{P}, \mathbb{Q}) = \sup_{\phi \in \Phi} \mathbb{E}_{\mathbb{Q}_n} \phi - \mathbb{E}_{\mathbb{P}_n} f^*(\phi)$.

**Approximation of the likelihood ratio**: An estimate of the likelihood ratio can be directly obtained from the variational approximation of $f$-divergences. The key observation is the following: *equality on Equation* (14) *is achieved iff* $\phi = \partial f\left(\frac{q_0}{p_0}\right)$. This tells us that the optimal solution to the variational approximation provides us with an estimator of the composite function $\partial f\left(\frac{q_0}{p_0}\right)$ of the likelihood ratio $\frac{q_0}{p_0}$. As long as we can invert $\partial f$, we can uniquely determine the likelihood ratio.

In the following, we shall get an empirical estimator of the likelihood ratio in two separate steps. We first solve the following:

$$\hat{\phi}_n := \arg\max_{\phi \in \Phi} \mathbb{E}_{\mathbb{Q}_n} \phi - \mathbb{E}_{\mathbb{P}_n} f^*(\phi) \tag{15}$$

which returns an estimator of $\partial f\left(\frac{q_0}{p_0}\right)$, not the ratio itself. And then we apply the inverse of $\partial f$ to $\hat{\phi}_n$. We therefore have a family of estimation methods for the likelihood function by simply ranging over choices of $f$.

**Remark**: If $f$ is not differentiable, then we cannot invert $\partial f$ but we can obtain estimators of other functions of the likelihood ratio. For instance, we can obtain an estimate of the thresholded likelihood ratio by using a convex function whose subgradient is the sign function centered at 1.

### A.4.1 LIKELIHOOD RATIO ESTIMATION WITH PEARSON DIVERGENCE

In this section, we show how to estimate the likelihood ratio when the Pearson divergence is used. With this choice, many computations simplify and we can write the estimator of the likelihood ratio in closed form. Other choices (such as the Kullback-Leibler divergence) are possible and legitimate, but usually do not lead to closed form expressions (see (Nguyen et al., 2010)).

The Pearson, or $\chi^2$, divergence is defined by the following choice: $f(t) := \frac{(t-1)^2}{2}$. The associated convex dual function is :

$$f^*(v) = \sup_{u \in \mathbb{R}} \left\{ uv - \frac{(u-1)^2}{2} \right\} = \frac{v^2}{2} + v.$$

Therefore the Pearson divergence is characterized by the following:

$$PE(\mathbb{P}||\mathbb{Q}) := \int p_0 \left( \frac{q_0}{p_0} - 1 \right)^2 d\mu \geq \sup_{\phi \in \Phi} \mathbb{E}_{\mathbb{Q}} \phi - \frac{1}{2} \mathbb{E}_{\mathbb{P}} \phi^2 - \mathbb{E}_{\mathbb{P}} \phi. \tag{16}$$

Solving the lower bound for the optimal $\phi$ provides us an estimator of $\partial f(\frac{q_0}{p_0}) = \frac{q_0}{p_0} - 1$. For the special case of the Pearson divergence, we can apply a change of variables which preserves convexity of the variational optimization problem Equation (16) and provides a more straightforward interpretation. Let the new variable be $z := \phi + 1$ with $z \in \mathcal{Z}$, which in this case is nothing but the inverse function of $\partial f$. We get

$$\sup_{\phi \in \Phi} \mathbb{E}_{\mathbb{Q}} \phi - \frac{1}{2} \mathbb{E}_{\mathbb{P}} \phi^2 - \mathbb{E}_{\mathbb{P}} \phi = \sup_{z \in \mathcal{Z}} \mathbb{E}_{\mathbb{Q}} z - \frac{1}{2} \mathbb{E}_{\mathbb{P}} z^2 - \frac{1}{2} \tag{17}$$

It is now trivial to see that $z$ is a 'direct' approximator of the likelihood ratio (i.e., it does not estimate a composite map of the likelihood ratio). Therefore for simplicity, we shall employ

$$\arg\min_{\phi \in \Phi} \frac{1}{2} \mathbb{E}_{\mathbb{P}} \phi^2 - \mathbb{E}_{\mathbb{Q}} \phi \tag{18}$$

to build our 'direct' estimator of the likelihood ratio.

Let the samples from $\mathbb{P}$ and $\mathbb{Q}$ be, respectively, $x_p(i)$ with $i = 1, ..., n_p$ and $x_q(i)$ with $i = 1, ..., n_q$. We define the empirical estimator of the likelihood ratio $\hat{\phi}_n$:

$$\hat{\phi}_n = \arg\min_{\phi \in \Phi} \frac{1}{2n_p} \sum_{i=1}^{n_p} \phi(x_p(i))^2 - \frac{1}{n_q} \sum_{i=1}^{n_q} \phi(x_f(i)). \tag{19}$$

**A closed form solution**: Up to now we have not defined in which class of functions our approximator $\phi$ lives. As done in (Nguyen et al., 2010; Liu et al., 2012), we choose $\phi \in \Phi$ where $\Phi$ is a RKHS induced by the kernel $k$.

We exploit the representer theorem to write a general function within $\Phi$ as:

$$\phi(x) = \sum_{i=1}^{n_{tr}} k(x, x_{tr}(i)) \alpha_i,$$

where we use $n_{tr}$ data which are the centers of the kernel sections used to approximate the unknown likelihood ratio (how to choose these centers is important and determines the approximation properties of $\hat{\phi}_n$). For now, we do not specify which data should be used as centers (we can use either data from $\mathbb{P}_n$ or from $\mathbb{Q}_n$ or from both or simply use user specified locations).

Let us define the following kernel matrices: $K_p := K(X_p, X_{tr}) \in \mathbb{R}^{n_p \times n_{tr}}$, $K_q := K(X_q, X_{tr}) \in \mathbb{R}^{n_q \times n_{tr}}$, where $X_p := \{x_p(i)\}$, $X_q := \{x_q(i)\}$ and $X_{tr} := \{x_{tr}(i)\}$.

We therefore have:

$$\hat{\phi}_n = \arg\min_{\phi \in \Phi} \frac{1}{2n_p} \sum_{i=1}^{n_p} \phi(x_p(i))^2 - \frac{1}{n_q} \sum_{i=1}^{n_q} \phi(x_f(i))$$

$$= \arg\min_{\alpha, \alpha \geq 0} \frac{1}{2n_p} \sum_{i=1}^{n_p} \left( \sum_{j=1}^{n_{tr}} k(x_p(i), x_{tr}(j)) \alpha_j \right)^2 - \frac{1}{n_f} \sum_{i=1}^{n_f} \sum_{j=1}^{n_{tr}} k(x_f(i), x_{tr}(j)) \alpha_j$$

$$= \arg\min_{\alpha, \alpha \geq 0} \frac{1}{2n_p} \alpha^T K_p^T K_p \alpha - \frac{1}{n_f} \mathbb{1}^T K_f \alpha$$

**Remark**: We impose the recombination coefficients $\alpha$ to be non negative since the likelihood ratio is a non negative quantity. The resulting optimization problem is a standard convex optimization problem with linear constraints which can be efficiently solved with Newton methods, nonetheless in general it does not admit any closed form solution.

We now relax the positivity constraints so that the optimal solution can be obtained in closed form. Moreover we add a quadratic regularization term as done in (Nguyen et al., 2010) which lead us to the following regularized optimization problem:

$$\arg\min_{\alpha} \frac{1}{2n_p} \alpha^T K_p^T K_p \alpha - \frac{1}{n_f} \mathbb{1}^T K_f \alpha + \frac{\gamma}{2} \|\alpha\|_{\Phi}^2$$

whose solution is trivially given by:

$$\hat{\alpha} = \frac{n_p}{n_f} \left( K_p^T K_p + n_p \gamma I_{n_{tr}} \right)^{-1} K_f^T \mathbb{1} := \frac{n_p}{n_f} H^{-1} K_f^T \mathbb{1} \tag{20}$$

The estimator of the likelihood ratio for an arbitrary location $x$ is given by the following:

$$\frac{p_q(x)}{p_p(x)} \approx \hat{\phi}_n(x) = K(x, X_{tr})\hat{\alpha} = \frac{n_p}{n_f} K(x, X_{tr}) \left( K_p^T K_p + n_p \gamma I_{n_{tr}} \right)^{-1} K_f^T \mathbb{1} \tag{21}$$

**Remark**: In the following we shall exploit RBF kernels which are defined by the length scales $\sigma$.

## A.5   SUBSPACE LIKELIHOOD RATIO ESTIMATION AND CUMSUM

In this section we describe our subspace likelihood ratio estimator and its relation to the CUMSUM algorithm. The CUMSUM algorithm requires to compute the likelihood ratio $\frac{p_f(y(t)|Y_c^{t-1})}{p_p(y(t)|Y_c^{t-1})}$ for each time $t$. We denote $p_p$ as the normal density and $p_f$ as the abnormal one (after the anomaly has occurred).

We shall proceed to express the conditional probability $p(y(t) \mid Y_1^{t-1})$ using our predictor. In particular it is always possible to express the optimal (unknown) one-step ahead predictor as:

$$\hat{y}_{t|t-1} = F^*(Y_{t-K+1}^t) := \mathbb{E}[y(t) \mid Y_{t-K+1}^t] \tag{22}$$

which is a deterministic function given the past of the time series (whose length is $K$). So that the data density distribution can be written in innovation form (based on the optimal prediction error) as:

$$y(t) = F^*(Y_{t-K+1}^t) + e(t) \tag{23}$$

where $e(t) := y(t) - F^*(Y_{t-K+1}^t)$ is, by definition, the one step ahead prediction error (or *innovation sequence*) of $y(t)$ given its past. We therefore have: $p(y(t) \mid Y_{t-K+1}^t) = p(e(t) \mid Y_{t-K+1}^t)$. Where $e(t)$ is the optimal prediction error for each time $t$ and is therefore indipendent on each time $t$.

**Remark**: In practice we do not know $F^*$ and we use our predictor learnt from normal data as a proxy. This implies the prediction residuals are approximately independent on normal data (the predictor can explain data well), while the prediction residuals are, in general, correlated on abnormal data.

To summarize: under normal conditions the joint distribution of $Y_c^t$ can be written as:

$$p(Y_c^t) = \prod_{i=c}^{t} p(y(i) \mid Y_c^{i-1}) = \prod_{i=c}^{t} p(e(i)) \qquad \text{normal conditions} \tag{24}$$

$$p(Y_c^t) = \prod_{i=c}^{t} p(y(i) \mid Y_c^{i-1}) = \prod_{i=c}^{t} p(e(i) \mid E_c^{i-1}) \qquad \text{abnormal conditions} \tag{25}$$

These two conditions in turn influence the log likelihood ratio test as follows: under $H_0 \implies \prod_{i=c}^{t} \frac{p_f(e(i))}{p_p(e(i))}$ while under $H_c \implies \prod_{i=c}^{t} \frac{p_f(e(i)|E_c^{i-1})}{p_p(e(i))}$. The main issue here is the numerator under $H_c$: the distribution of residuals changes at each time-stamp (it is a conditional distribution) and $p_f(e(i) \mid E_c^{i-1})$ is difficult to approximate (it requires the model of the fault). In the following we show that replacing $p_f(e(i) \mid E_c^{i-1})$ with $p_f(e(i))$ allows us to compute a

lower bound on the cumulative sum. Such an approximation is necessary to estimate the likelihood ratio in abnormal conditions, the main downside of this approximation is that the detector becomes slower (it needs more time to reach the stopping time threshold).

**Applying the independent likelihood test in a correlated setting**: We now show that treating $p_f(e(i) \mid E_c^{i-1})$ as independent random variables $p_f(e(i))$ for $i = 1, ..., t$ allows us to compute a lower bound on the log likelihood $\log \Omega_c^t$ (i.e. the cumulative sum). We denote the cumulative sum of the log likelihood ratio using independent variables as $\log \bar{\Omega}_c^t = \sum_{i=c}^t \log \frac{p_f(e(i))}{p_p(e(i))}$

**Proposition A.2.** *Assume a change happens at time $c$ so that $H_c$ is true and the following log likelihood ratio holds true:* $\log \Omega_c^t = \sum_{i=c}^t \log \frac{p_f(e(i)|E_c^{i-1})}{p_p(e(i))}$. *Then it holds* $\log \Omega_c^t \geq \log \bar{\Omega}_c^t$.

*Proof.* By simple algebra we can write:

$$\frac{p_f(e(i) \mid E_c^{i-1})}{p_p(e(i))} = \frac{p_f(e(i) \mid E_c^{i-1})}{p_f(e(i))} \frac{p_f(e(i))}{p_p(e(i))} \qquad \forall i$$

Now recall the cumulative sum of the log-likelihood ratios taken under the current data generating mechanism $p_f(E_1^t)$ provides an estimate of the expected value of the log-likelihood ratio. Due to the correlated nature of data $E_1^t$ the samples are drawn from a multidimensional distribution of dimension $t$ (a sample from this distribution is an entire trajectory from $c$ to $t$).

We now take the expectation of previous formula w.r.t. the 'true' distribution $p_f(E_1^t)$:

$$\mathbb{E}_{p_f(E_c^t)} \Omega_c^t = \mathbb{E}_{p_f(E_c^t)} \log \prod_{i=c}^t \frac{p_f(e(i) \mid E_c^{i-1})}{p_p(e(i))}$$

$$= \mathbb{E}_{p_f(E_c^t)} \log \prod_{i=c}^t \frac{p_f(e(i) \mid E_c^{i-1})}{p_f(e(i))} + \mathbb{E}_{p_f(E_c^t)} \log \prod_{i=c}^t \frac{p_f(e(i))}{p_p(e(i))}$$

$$= MI\Big(p_f(E_c^t); \prod_{i=c}^t p_f(e(i))\Big) + KL\Big(\prod_{i=c}^t p_f(e(i)) \Big\| \prod_{i=c}^t p_p(e(i))\Big)$$

$$\geq KL\Big(\prod_{i=c}^t p_f(e(i)) \Big\| \prod_{i=c}^t p_p(e(i))\Big)$$

where we used the fact the mutual information is always non negative. $\qquad \square$

**How to approximate pre and post fault distributions**: Both $p_p$ and $p_f$ are not known and their likelihood ratio need to be estimated from available data. From Section 5.1.1 we know how to approximate the likelihood ratio given two set of data without estimating the densities. In our anomaly detection setup we define these two sets as: $E_p := E_{t-n_p-n_f+1}^{t-n_f}$ and $E_f := E_{t-n_f+1}^t$. So that given current time $t$ we look back at a window of length $n_p + n_f$. The underlying assumption is that under $H_c$ normal data are present in $E_p$ and abnormal ones in $E_f$. We estimate the likelihood ratio $\frac{p_f(e(t))}{p_p(e(t))}$ at each time $t$ by assuming both $E_p$ and $E_f$ data are independent (see Proposition A.2) and cumulate their log as $t$ increases.

**How do $n_p$ and $n_f$ affect our detector?** The choice of the windows length ($n_p$ and $n_f$) is fundamental and highly influences the likelihood estimator. Using small windows makes the detector highly sensible to both point and sequential outliers, while larger windows are better suited to estimate only sequential outliers. We now assume $n_p = n_f$ and study how small and large values affect the behaviour of our detector in simple working conditions.

In Figure 5 and Figure 6 we compute the cumulative sum of log likelihood ratios estimated from data on equally sized windows. Intuitively any local minimum after a 'large' (depending on the threshold $\tau$) increase of the cumulative sum is a candidate abnormal point.

In Figure 7 and Figure 8 we compare the cumulative sum of estimated likelihood ratios on data in which both sequential and point outliers are present. In particular we highlight that large window sizes $n_p$ and $n_f$ are usually not able to capture point anomalies Figure 7 while using small window sizes allow to detect both (at the expenses of a more sensitive detector) Figure 8.

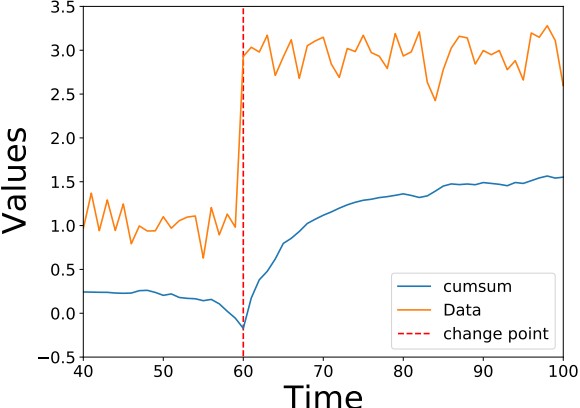

Figure 5: **Change point**: Cumulative sum (blue) obtained with our method in a synthetic example. We use the cumulative sum of estimated likelihood ratios on data in which a change point is present at $t = 60$. We use $n_p = n_f = 20$ and kernel length scale=0.2

Figure 6: **Point anomaly**: Cumulative sum (blue) obtained with our method in a synthetic example. We use the cumulative sum of estimated likelihood ratios on data in which a point outlier is present at $t = 60$. We use $n_p = n_f = 2$ and kernel length scale=2.

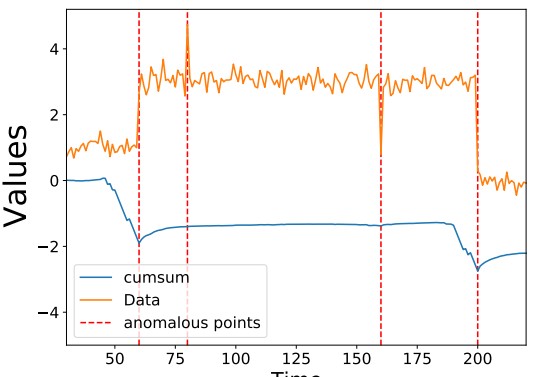

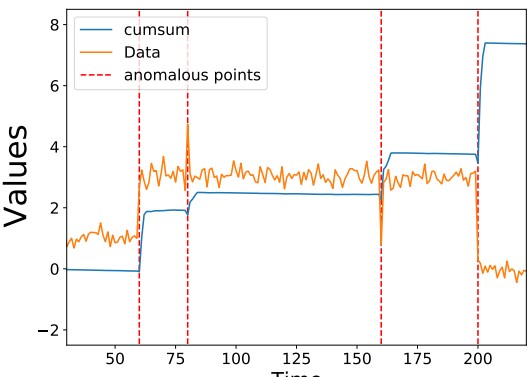

Figure 7: **Large $n_p$ and $n_f$**: Cumulative sum (blue) obtained with our method in a synthetic example. We use the cumulative sum of estimated likelihood ratios on data which contain both change points ($t = 60$ and $t = 200$) and point outliers ($t = 80$ and $t = 160$). We use $n_p = n_f = 20$ and kernel length scale=1.

Figure 8: **Small $n_p$ and $n_f$**: Cumulative sum (blue) obtained with our method in a synthetic example. We use the cumulative sum of estimated likelihood ratios on data which contain both change points ($t = 60$ and $t = 200$) and point outliers ($t = 80$ and $t = 160$). We use $n_p = n_f = 3$ and kernel length scale=5.

## A.6 DATASETS

### A.6.1 YAHOO DATASET

Yahoo Webscope dataset (Laptev & Amizadeh, 2020) is a publicly available dataset containing 367 real and synthetic time series with point anomalies, contextual anomalies and change points. Each time series contains 1420-1680 time stamps. This dataset is further divided into 4 sub-benchmarks: A1 Benchmark, A2 Benchmark, A3 Benchmark and A4 Benchmark. A1Benchmark is based on the real production traffic to some of the Yahoo! properties. The other 3 benchmarks are based on synthetic time series. A2 and A3 Benchmarks include point outliers, while the A4Benchmark includes change-point anomalies. All benchmarks have labelled anomalies. We use such information only during evaluation phase (since our method is completely unsupervised).

### A.6.2 NAB DATASET

NAB (Numenta Anomaly Benchmark) (Lavin & Ahmad, 2015) is a publicly available anomaly detection benchmark. It consists of 58 data streams, each with 1,000 - 22000 instances. This dataset contains streaming data from different domains including read traffic, network utilization, on-line advertisement, and internet traffic. As done in (Geiger et al., 2020) we choose a subset of NAB benchmark, in particular we focus on the NAB Traffic and NAB Tweets benchmarks.

### A.6.3 CO2 DATASET

We test the prediction and interpretability capabilities of our model on the CO2 dataset from kaggle [2]. The main goal here is to predict both trend and periodicity of CO2 emission rates on different years. Note this is not an Anomaly detection task.

Table 3: **Datasets summaries**. We report some properties of the datasets used (see (Geiger et al., 2020) for mode details).

| | | Yahoo | | | NAB | | Kaggle |
|---|---|---|---|---|---|---|---|
| | A1 | A2 | A3 | A4 | Traffic | Tweets | CO2 |
| # signals | 67 | 100 | 100 | 100 | 7 | 10 | 9 |
| # anomalies | 178 | 200 | 939 | 835 | 14 | 33 | |
| point | 68 | 33 | 935 | 833 | 0 | 0 | |
| sequential | 110 | 167 | 4 | 2 | 14 | 33 | |
| # anomalous points | 1669 | 466 | 943 | 837 | 1560 | 15651 | |
| (% tot) | 1.8% | 0.32% | 0.56% | 0.5% | 9.96% | 9.87% | |
| # data points | 94866 | 142100 | 168000 | 168000 | 15662 | 158511 | 4323 |

### A.6.4 NYT DATASET

The New York Times Annotated Corpus[3] (Sandhaus, 2008) contains over 1.8 million articles written and published by the New York Times between January 1, 1987 and June 19, 2007. We pre-processed the lead paragraph of each article with a pre-trained BERT model (Devlin et al., 2019) from the HuggingFace Transformers library (Wolf et al., 2020) and extracted the 768-dimensional hidden state of the [CLS] token (which serves as an article-level embedding). For each day between January 1, 2000 and June 19, 2007, we took the mean of the embeddings of all articles from that day. Finally, we computed a PCA and kept the first 200 principal components (which explain approximately 95% of the variance), thus obtaining a 200-dimensional time series spanning 2727 consecutive days. Note that we did not use any of the dataset's annotations, contrary to prior work such as Rayana & Akoglu (2015).

### A.7 EXPERIMENTAL SETUP

In this section, we shall describe the experimental setup we used to test STRIC.

**Data preprocessing**: Before learning the predictor we standardize each dataset to have zero mean and standard deviation equals to one. As done in (Braei & Wagner, 2020) we note standardization is not equal to normalization, where data are forced to belong to the interval $(0, 1)$. Normalization is more sensitive to outliers, thus it would be inappropriate to normalize our datasets, which contain outliers.

We do not apply any deseasonalizing or detrending pre-processing.

**Data splitting**: We split each dataset into training and test sets preserving time ordering, so that the first data are used as train set and the following ones are used as test set. The data used to validate the model during optimization are last $10\%$ of the training dataset. Depending on the experiment, we choose a different percentage in splitting train and test. When comparing with (Braei & Wagner, 2020) we used $30\%$ as training data, while when comparing to (Munir et al., 2019) we use $40\%$. Such a choice is dictated by the particular (non uniform) experimental setup reported in (Braei &

---

[2]https://www.kaggle.com/txtrouble/carbon-emissions
[3]https://catalog.ldc.upenn.edu/LDC2008T19

Wagner, 2020; Munir et al., 2019) and has been chosen to produce comparable results with state of the art methods present in literature.

**Evaluation metrics**: We compare different predictors by means of the RMSE (root mean squared error) on the one-step ahead prediction errors. Given a sequence of data $Y_1^N$ and the one-step ahead predictions $\hat{Y}_1^N$ the RMSE is defined as: $\sqrt{\frac{1}{N} \sum_{i=1}^{N} \|y(i) - \hat{y}(i)\|^2}$.

As done in (Braei & Wagner, 2020) we compare different anomaly detection methods taking into account several metrics. We use F1-Score which is defined as the armonic mean of Precision and Recall (see (Braei & Wagner, 2020; Munir et al., 2019)) and another metric that is often used is *receiver operating characteristic curve*, ROC-Curve, and its associated metric *area under the curve* (AUC). The AUC is defined as the area under the ROC-Curve. This metric is particularly useful in our anomaly detection setting since it describes with an unique number *true positive rate* and *false positive rate* on different threshold values. We now follow (Braei & Wagner, 2020) to describe how AUC is computed. Let the *true positive rate* and *false positive rate* be defined, respectively, as: $TPR = \frac{TP}{P}$ and $FPR = \frac{FP}{N}$, where $TP$ stands for *true positive*, $P$ for *positive*, $FP$ for *false positive* and $N$ for *negative*. To copute the ROC-Curve we use different thresholds on our anomaly detection method. We therefore have different pairs of $TPR$ and $FPR$ for each threshold. These values can be plotted on a plot whose $x$ and $y$ axes are, respectively: $FPR$ and $TPR$. The resulting curve starts at the origin and ends in the point (1,1). The AUC is the area under this curve. In anomaly detection, the AUC expresses the probability that the measured algorithm assigns a random anomalous point in the time series a higher anomaly score than a random normal point.

**Hardware**: We conduct our experiments on the following hardware setup:

- Processor: Intel(R) Core(TM) i9-10980XE CPU @ 3.00GHz
- RAM: 128 Gb
- GPU: Nvidia TITAN V 12Gb and Nvidia TITAN RTX 24Gb

**Hyper-parameters**: All the experiments we carried out are uniform on the optimization hyper-parameters. In particular we fixed the maximum number of epochs to 300, the learning rate to 0.001 and batch size to 100. We optimize each model using Adam and early stopping.

We fix STRIC's first module hyper-parameters as follows:

- number of filter per block: $l_0 = 10$, $l_1 = 100$, $l_2 = 200$
- linear filters kernel lengths ($N_0$, $N_1$, $N_2$): half predictor's memory

In all experiments we either use a TCN composed of 3 hidden layers with 300 nodes per layer or a TCN with 8 layers and 32 nodes per layer. Moreover we chose $N_3 = 5$ (TCN kernels' lengths) and relu activation functions (Bai et al., 2018).

**Comparison with SOTA methods:** We tested our model against other SOTA methods (Table 1) in a comparable experimental setup. In particular, we chose comparable window lengths and architecture sizes (same order of magnitude of the number of parameters) to make the comparison as fair as possible. For the hyper-parameters details of any SOTA method we used we refer the relative cited reference. We point out that while the window length is a critical hyper-parameter for the accuracy of many methods, our architecture is robust w.r.t. choice of window length: thanks to our fading regularization, the user is required only to choose a window length larger than the optimal one and then our automatic complexity selection is guaranteed to find the optimal model complexity given the available data Section 4.1.

**Anomaly scores**: When computing the F-score we use the predictions of the CUMSUM detector which we collect as a binary vector whose length is the same as the number of available data. Ones are associated to the presence of an anomalous time instants while zeros are associated to normality.

When computing the AUC we need to consider a continuous anomaly score, therefore the zero-one encoded vector from the CUMSUM is not usable. We compute the anomaly scores for each time instant as the estimated likelihood ratios. Since we write the likelihood ratio as $\frac{p_f}{p_p}$, it is large when data does not come from $p_p$ (which we consider the reference distribution).

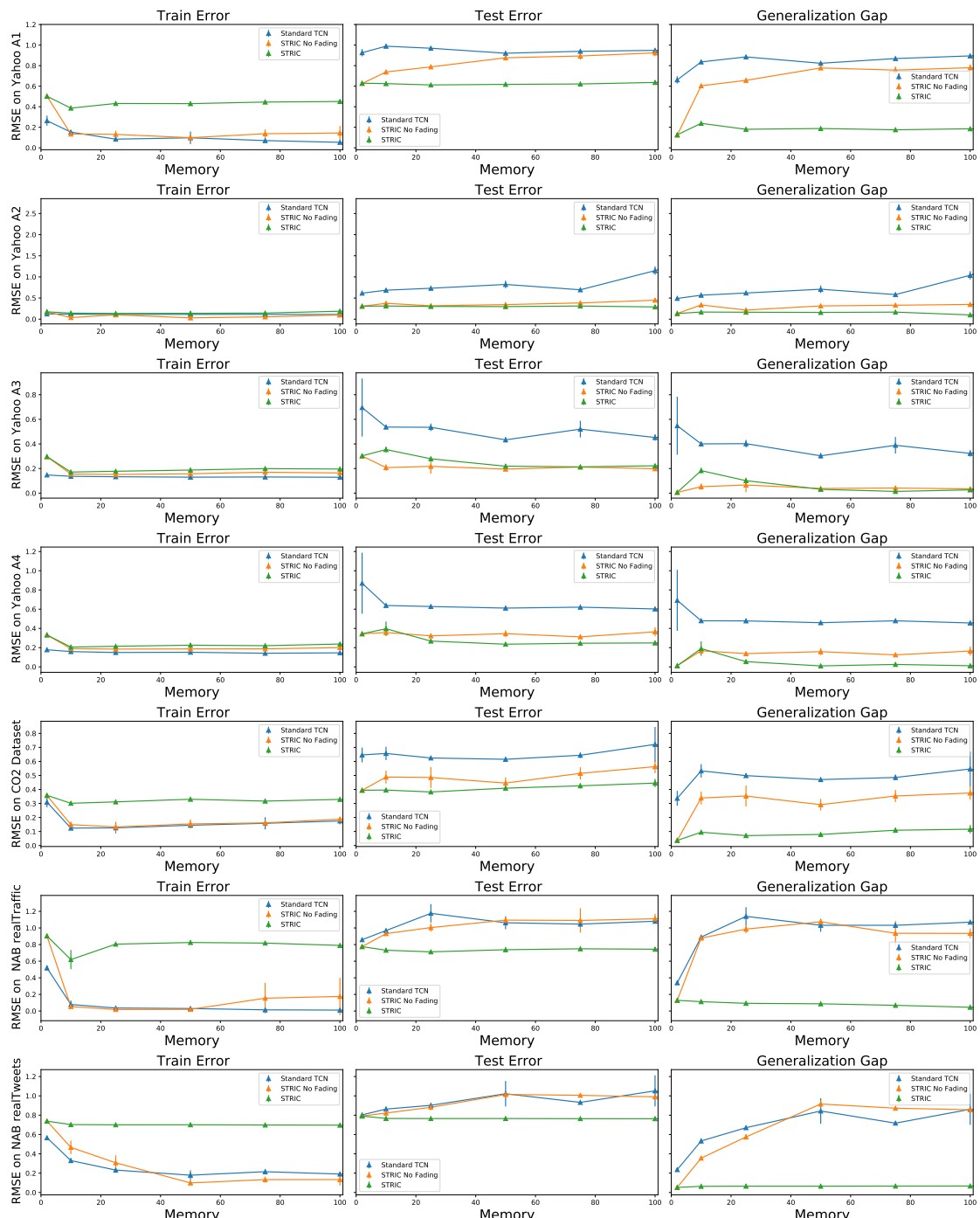

Figure 9: **Ablation studies on different datasets**: Effects of interpretable blocks and fading regularization on model's forecasting as the available window of past data increases (memory). **Left Panel**: Train error. **Center Panel**: Test error. **Right Panel**: Generalization Gap. The test error of STRIC is uniformiy smaller than a standard TCN (without interpretable blocks nor fading regularization). Adding interpretable blocks to a standard TCN improves generalization for fixed memory w.r.t. Standard TCN but get worse (overfitting occurs) as soon as the available past data horizon increase. Fading regularization is effective: STRIC generalization GAP is almost constant w.r.t. past horizon.

### A.7.1 ABLATION STUDY

In Figure 9 we show different metrics based on the predictor's RMSE (training, test and generalization gap) as a function of the memory of the predictor. We test our fading regularization on a variety of different datasets. In all situations fading regularization helps improving test generalization and preserving the generalization gap (by keeping it constant) as the model complexity increases. All plots show confidence intervals around mean values evaluated on 10 different random seeds.

In Table 4 we extend the results we show in the main paper by adding uncertainties (measured by standard deviations on 10 different random seeds) to the values of train and test RMSE on different ablations of STRIC. Despite the high variability across different datasets STRIC achieves the most consistent results (smaller standard deviations both on training and testing).

Finally, in Table 5 we show the effects on different choices of the predictor's memory $n_{pred}$ and length of the anomaly detectors windows $n_{det}$ on the detection performance of STRIC. Note both F-score and AUC are highly sensible to the choice of $n_{det}$: the best results are achieve for small windows. On the other hand when $n_{det}$ is large the performance drops. This is due to the type of anomalies present in the Yahoo benchmark: most of the them can be considered to be point anomalies. In fact, as we showed in Appendix A.5, our detector is less sensible to point anomalies when a large window $n_{det}$ is chosen.

In Table 5 we also report the reconstruction error of the optimal predictor given it's memory $n_{pred}$. Note small memory in the predictor introduce modelling bias (higher training error) while a large memory does not (thanks to fading regularization). As we observed in Appendix A.5 better predictive models provide the detection module with more discriminative residuals: the downstream detection module achieves better F-scores and AUC.

Table 4: **Ablation study on the RMSE of prediciton errors with standard deviation on 10 different seeds**: We compare a standard TCN model with our STRIC predictor and some variation of it (using the same train hyperparameters). The effect of adding a linear interpretable model before a TCN improves generalization error. Fading regularization has a beneficial effect in controlling the complexity of the TCN model and reducing the generalization gap.

| | | TCN | | TCN + Linear | | TCN + Fading | | STRIC pred | |
|---|---|---|---|---|---|---|---|---|---|
| | | Train | Test | Train | Test | Train | Test | Train | Test |
| Datasets | Yahoo A1 | **0.10** ± 0.06 | 0.92 ± 0.06 | **0.10** ± 0.03 | 0.88 ± 0.03 | 0.44 ± 0.03 | 0.92 ± 0.03 | 0.43 ± 0.02 | **0.62** ± 0.02 |
| | Yahoo A2 | **0.11** ± 0.02 | 0.82 ± 0.02 | 0.13 ± 0.01 | 0.35 ± 0.02 | 0.20 ± 0.01 | 0.71 ± 0.01 | 0.14 ± 0.01 | **0.30** ± 0.01 |
| | Yahoo A3 | **0.13** ± 0.01 | 0.43 ± 0.01 | 0.16 ± 0.01 | **0.22** ± 0.01 | 0.15 ± 0.01 | 0.40 ± 0.01 | 0.19 ± 0.01 | **0.22** ± 0.01 |
| | Yahoo A4 | **0.15** ± 0.01 | 0.61 ± 0.01 | 0.19 ± 0.01 | 0.35 ± 0.01 | 0.17 ± 0.01 | 0.55 ± 0.01 | 0.23 ± 0.01 | **0.24** ± 0.01 |
| | CO2 Dataset | **0.14** ± 0.02 | 0.62 ± 0.02 | 0.15 ± 0.02 | 0.45 ± 0.02 | 0.18 ± 0.03 | 0.61 ± 0.03 | 0.33 ± 0.01 | **0.41** ± 0.01 |
| | NAB Traffic | **0.03** ± 0.01 | 1.06 ± 0.02 | 0.04 ± 0.01 | 1.00 ± 0.02 | 0.62 ± 0.01 | 0.93 ± 0.01 | 0.63 ± 0.02 | **0.74** ± 0.02 |
| | NAB Tweets | **0.18** ± 0.05 | 1.02 ± 0.05 | 0.20 ± 0.05 | 0.98 ± 0.05 | 0.47 ± 0.02 | 0.83 ± 0.02 | 0.70 ± 0.01 | **0.77** ± 0.01 |

Table 5: **Sensitivity of STRIC to hyper-parameters**: We compare STRIC on different anomaly detection benchmarks datasets using different hyper-parameters: memory of the predictor $n_{pred}$ and length of anomaly detector windows $n_p = n_f = n_{det}$.

| | | Yahoo A1 | | Yahoo A2 | | Yahoo A3 | | Yahoo A4 | |
|---|---|---|---|---|---|---|---|---|---|
| | | F1 | AUC | F1 | AUC | F1 | AUC | F1 | AUC |
| Models | $n_{pred} = 10, n_{det} = 2$ | 0.45 | 0.89 | 0.63 | 0.99 | 0.87 | 0.99 | 0.64 | 0.89 |
| | $n_{pred} = 100, n_{det} = 2$ | **0.48** | **0.9308** | **0.98** | **0.9999** | **0.99** | **0.9999** | **0.68** | **0.9348** |
| | $n_{pred} = 10, n_{det} = 20$ | 0.10 | 0.58 | 0.63 | 0.99 | 0.47 | 0.83 | 0.37 | 0.72 |
| | $n_{pred} = 100, n_{det} = 20$ | 0.10 | 0.55 | **0.98** | **0.9999** | 0.49 | 0.86 | 0.35 | 0.76 |
| | | Yahoo A1 | | Yahoo A2 | | Yahoo A3 | | Yahoo A4 | |
| | | Train | Test | Train | Test | Train | Test | Train | Test |
| Models | $n_{pred} = 10$ | 0.44 | 0.62 | 0.16 | 0.31 | 0.22 | 0.23 | 0.25 | 0.26 |
| | $n_{pred} = 100$ | 0.42 | 0.61 | 0.14 | 0.30 | 0.19 | 0.22 | 0.23 | 0.24 |

### A.7.2 COMPARISON TCN VS STRIC

In this section we show standard non-linear TCN without regularization and proper inductive bias might not generalize on non-stationary time series (e.g. time series with non zero trend component) and TCN architecture. In Figure 10 we

compare the prediciton errors of a standard TCN model against our STRIC on the A3 Yahoo dataset. We train both models using the same optimization hyper-parameters (as described in previous section). Note a plain TCN does not necessarily capture the trend component in the test set.

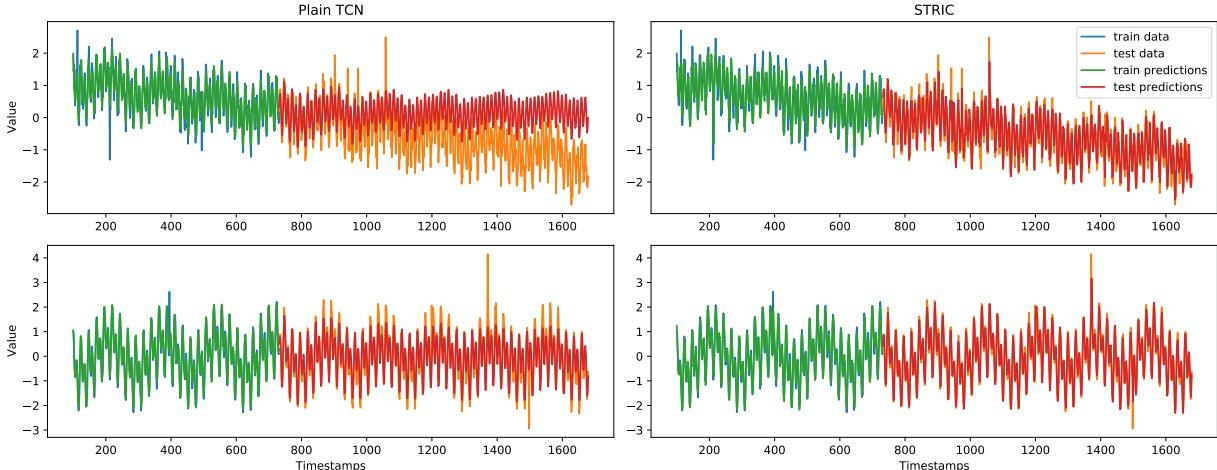

Figure 10: We compare an off-the-shelf TCN against STRIC (time series predictor) on the Yahoo dataset A3 Benchmark. Note the standard TCN overfits compared to STRIC: the standard TCN does not handle correctly the trend component of the signal (**First row**). If we consider a time series without trend, the standard TCN model performs better but overfitting is still present. In particular the generalization gap (measured using squared reconstruction error) for the two models is: Standard TCN 0.3735 and STRIC 0.0135.

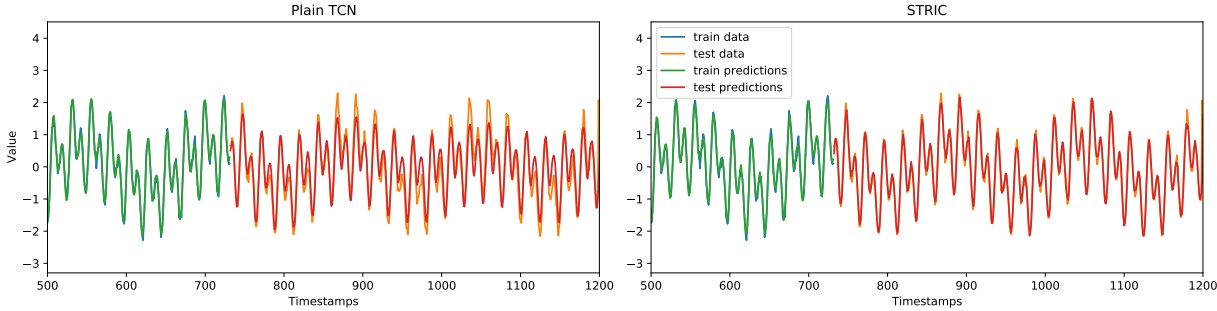

Figure 11: Zoom on the second row of panels in Figure 10. We show the interface between train and test data both on a plain TCN and on our STRIC predictor. A plain TCN overfits w.r.t. STRIC also when not trend is present.

## A.8  STRIC vs SOTA Anomaly Detectors

In this section we further expand the discussion on the main differences between STRIC and other SOTA anomaly detectors by commenting results obtained in Table 1 and Table 6. Table 6 is used to highlight the relative performance of STRIC when the peformance are nearly saturated (e.g. Yahoo A2 and A3): we report the relative performance indeces against TRAID for all the models we tested in Table 6. Both for F1 and AUC we report the following for each comparing SOTA method: $\frac{\text{AUC}_{\text{method}} - \text{AUC}_{\text{STRIC}}}{1 - \text{AUC}_{\text{STRIC}}} \cdot 100$ (similarly for F1).

To begin with, STRIC outperforms 'traditional' methods (LOF and One-class SVM) which are considered as baselines models for comparing time series anomaly detectors.

Table 6: **Comparison with SOTA anomaly detectors:** We compare STRIC with other anomaly detection methods on the experimental setup and the same evaluation metrics proposed in (Braei & Wagner, 2020; Munir et al., 2019). The baseline models are: MA, ARIMA, LOF (Shen et al., 2020), LSTM (Braei & Wagner, 2020; Munir et al., 2019), Wavenet (Braei & Wagner, 2020) , Yahoo EGADS (Munir et al., 2019) , GOAD (Bergman & Hoshen, 2020), OmniAnomaly (Su et al., 2019), Twitter AD (Munir et al., 2019), TanoGAN (Bashar & Nayak, 2020), TadGAN (Geiger et al., 2020) , DeepAR (Flunkert et al., 2017) and DeepAnT (Munir et al., 2019) . STRIC outperforms most of the other methods based on statistical models and based on DNNs. Same as Table 1, here we report the relative improvements w.r.t. STRIC (the higher the better).

| Relative F1-score improvement over STRIC in % | Yahoo A1 | Yahoo A2 | Yahoo A3 | Yahoo A4 | NAB Tweets | NAB Traffic |
|---|---|---|---|---|---|---|
| ARIMA | -20 | -88 | -42 | **6** | -33 | -37 |
| LSTM | -7 | -33 | -60 | - 21 | | |
| Yahoo EGADS | -1 | -95 | -78 | -54 | | |
| OmniAnomaly | -1 | -60 | -45 | -11 | -6 | -10 |
| Twitter AD | **0** | -98 | -85 | -53 | | |
| TanoGAN | -11 | -85 | -73 | -13 | -36 | -44 |
| TadGAN | -13 | -85 | -65 | -20 | -25 | -47 |
| DeepAR | -29 | -72 | -79 | -41 | -37 | -32 |
| DeepAnT | -4 | -67 | -15 | 0 | | |
| STRIC (ours) | **0** | **0** | **0** | 0 | **0** | **0** |

| Relative AUC improvement over STRIC in % | Yahoo A1 | Yahoo A2 | Yahoo A3 | Yahoo A4 | NAB Tweets | NAB Traffic |
|---|---|---|---|---|---|---|
| MA | -47 | -98 | -98 | **379** | | |
| ARIMA | -45 | -99 | -99 | 124 | | |
| LOF | -28 | -99 | -99 | -81 | -32 | -44 |
| Wavenet | -60 | -99 | -99 | -84 | | |
| LSTM | -63 | -99 | -99 | -84 | | |
| GOAD | -37 | -99 | -99 | -51 | -19 | -12 |
| DeepAnT | -32 | -99 | -99 | -53 | -23 | -13 |
| STRIC (ours) | **0** | **0** | **0** | 0 | **0** | **0** |

**Comparison with other Deep Learning based methods**: STRIC outperforms most of the SOTA Deep Learning based methods reported in Table 1: TadGAN, TAnoGAN, DeepAnT and DeepAR (the last one is a SOTA time series predictor). Note the relative improvement of STRIC is higher on the Yahoo dataset where statistical models outperforms deep learning based ones. We believe this is due to both fading regularization and the seasonal-trend decomposition performed by STRIC.

Despite the general applicability of GOAD (Bergman & Hoshen, 2020) this method has not been designed to handle time series, but images and tabular data. "Geometric" transformations which have been considered in GOAD and actually have inspired it (rotations, reflections, translations) might not be straightforwardly applied to time series. Nevertheless, while we have not been able to find in the literature any direct and principled extension of this work to the time series domain, we have implemented and compared against (Bergman & Hoshen, 2020) by extending the main design ideas of GOAD to time-series. So that we applied their method on lagged windows extracted from time series (exploiting the same architectures proposed for tabular data case with some minor modifications). We report the results we obtained by running the GOAD's official code on all our benchmark datasets. Overall, STRIC performs (on average) 70% better than GOAD on the Yahoo dataset and 15% better on the NAB dataset.

### A.8.1 DETAILS ON THE NYT EXPERIMENT

Figure 4 shows the normalized anomaly score computed by STRIC on the NYT dataset, following the setup described in Appendix A.6.4. Some additional insights can be gained by zooming in around some of the detected change-points. In Figure 12 (left), we see that the anomaly score (blue line) rapidly increases immediately after the 9/11 attack and reaches its peak some days later. Such delay is inherently tied to our choice of time scale, that privileges the detection of prolonged anomalies as opposed to single-day anomalies (which are not meaningful due to the high variability of

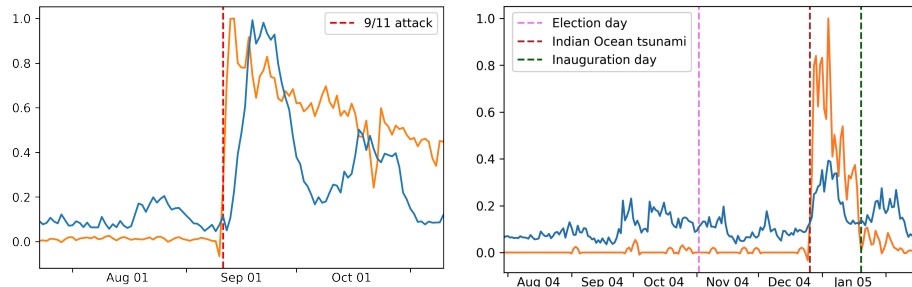

Figure 12: A closer look at some of the change-points detected by STRIC. **Left**: Normalized anomaly score (blue line) and normalized frequency of the "Terrorism" descriptor (orange line) around the 9/11 attack. **Right**: Normalized anomaly score (blue line) and normalized frequency of the "Earthquakes" descriptor (orange line) in the second half of 2004 and beginning of 2005. The 2004 U.S. election causes an increase in the anomaly score, but the most significant change-point occurs after the Indian Ocean tsunami.

the news content). The change-point which occurs the day after the 9/11 attack is reflected by a sudden increase of the relative frequency of article descriptors such as "Terrorism" (orange line). Article descriptors are annotated in the NYT dataset, but they are not given as input to STRIC so that we do not rely on any human annotations. However, they can help interpreting the change-points found by STRIC.

In Figure 12 (right), we can observe that the anomaly score (blue line) is higher in the months around the 2004 U.S. election and immediately after the inauguration day. However, the highest values for the anomaly score occur around the end of 2004, shortly after the Indian Ocean tsunami. Indeed, this is reflected by an abrupt increase of the frequency of descriptors like "Earthquakes" (orange line) and "Tsunami".

We note this experiment is qualitative and unfortunately we are not aware of any ground truth or metrics (e.g., in Rayana & Akoglu (2015) a similar qualitative result has been reported on the NYT dataset). We therefore tested STRIC against a simple baseline which uses PCA on BERT features and a threshold to detect anomalies. Despite being a simple baseline, this method prooved to be highly applied in practice due to its simplicity (Blázquez-García et al., 2020). The PCA + threshold baseline is able to pick up some events (2000 election, 9/11 attack, housing bubble) but is otherwise more noisy than STRIC's anomaly score. This is likely due to the lack of a modeling of seasonal/periodic components. For instance, the anomaly score of the simple baseline contains many false alarms which are related to normal weekly periodicity that is not easily modeled by the baseline. This does not affect STRIC's predictions, since normal weekly periodicity is directly modeled and identified as normal behaviour.

