# OpenReview forum: "STRIC: Stacked Residuals of Interpretable Components for Time Series Anomaly Detection"
_ICLR.cc/2022/Conference — ICLR 2022 Submitted_

### Official Review · Reviewer_rnBY · 2021-11-02

**Correctness:** 3
**Technical Novelty And Significance:** 3
**Empirical Novelty And Significance:** 2
**Recommendation:** 5
**Confidence:** 5

**Main Review:**

·	Strength:

o	The motivation of each components are clearly justified.

o	The empirical result on univariate time series data outperforms most of the baselines.

o	To the best of my knowledge, the Bayesian learning formulation and CUMSUM algorithm with subspace likelihood ratio estimation are novel.

·	Weakness:

o	Empirical evaluation is only conducted on univariate time series data, making it not convincing that it is capable of dealing with multivariate time series data (which is very common in many application scenarios). Add some evaluations on multivariate time series may help on eliminate the concern.

o	The selected datasets has been shown as flawed benchmark [1] and can be easily solved with few lines of rules. Conducting experiment on these datasets making it questionable about the performance of the proposed algorithm. Either adopting new datasets to evaluate the algorithm or involving the one-line rule as baselines may be helpful to eliminate the concern.

o	The proposed residual framework assume that the input time series is the addition of trend, seasonality and residual series. However, it is very likely that the three components are not linearly dependent to the resulting time series. Providing more justifications on using addition to model aggregate the information may be helpful to eliminate the concern.

o	There is no empirical evaluation of the interpretable components. The author only show the result of decomposition without illustrating what makes it helpful for interpreting outlier detection result (which is one of the main contributions of this paper). Adding more justification with outlier annotations on Figure 2 may be helpful for readers to understand the contribution in empirical analysis.

[1] Renjie Wu, Eamon Keogh, Current Time Series Anomaly Detection Benchmarks are Flawed and are Creating the Illusion of Progress, IEEE Transactions on Knowledge and Data Engineering, 2021


**Summary Of The Paper:**

·	An interpretable stacked residual framework is proposed for time series anomaly detection with interpretation on seasonality and trend of the target time series.

·	A novel regularization term is proposed to optimize network parameters with Bayesian Optimization

·	An extension of CUMSUM algorithm is proposed to detect outliers without knowledge of pre- and post-distributions of the data.


**Summary Of The Review:**

·	To summarize, this paper proposed a novel interpretable outlier detection algorithm with stack residual blocks, regularized model complexity selection and novel CUMSUM detection algorithm. The proposed method is technically sound. However, the selecting datasets are solely in univariate setting and are shown to be flawed in previous study, which makes the empirical evaluation less credible. In addition, technical justification and empirical analysis of residual framework is missing, making it hard to evaluate the contribution of interpretable components. I believe some more modifications will certainly improve the quality of paper to meet the ICLR standard. Therefore, I vote for marginally below the acceptance threshold.

---

> ### Author Response · Authors · 2021-11-22
> **New Benchmark Results & Interpretable Components discussion**
>
> We sincerely thank the Reviewer for the comments. Below we address all the comments.
>
> ## Benchmarks (reviewer’s point 1 and 2)
> To further address reviewer’s concerns we tested STRIC on a different anomaly detection benchmark [2] (as suggested by reviewer zX4p) which contains real world multivariate time series. In particular our results suggest STRIC outperforms any deep learning based method in [2] together with other SOTA alternatives: AR, IForest and OCSVM. Moreover, thanks to our automatic complexity selection, STRIC is less sensitive w.r.t. the length of the window of past values used to feed it. For these experiments we fixed a common length of 50 (for all datasets).
>
> In particular STRIC achieves:
> - 5% improvement over GBRT in the Credit Card Dataset
> - 10% improvement over LSTM-RNN in the CICIDS Dataset
> - 7% improvement over IForset in the GECCO Dataset
> - 13% improvement over IForset in the SWAN-SF Dataset
>
> ## STL-like decomposition assumptions (reviewer’s point 3)
> If a multiplicative decomposition is required it would simply be necessary to pre-process the time-series by means of a logarithmic function. This is a standard approach used for time series decomposition techniques (see [3] and [4]) and it is indeed applicable to our method. We point out that the current version of our algorithm does not automatically discover which type of decomposition to use and it is the user's responsibility to recognize whether the additive or multiplicative decomposition is better suited, this is not different to standard practice on STL based methods.
>
> Indeed our method resembles STL, but differently from STL it extracts seasonal and trend components jointly, while also optimizing an architecture that includes a non-linear model on top of the decomposition. Not only does this improve the final predictions and anomaly scores, but it also relieves the STL from having to model complex non-linear parts of the signal. Therefore, it results in a more accurate seasonal-trend decomposition than a standard STL model. We added a new figure in the appendix to show the improvement over standard STL decomposition.
>
> ## Interpretable components (reviewer’s point 4)
> We believe the empirical evaluation of the interpretable components should be splitted in two parts:
> - the effect of the interpretable module on the predictive model
> - the interpretation of the outlier detection results
>
> We answered the first point through our ablation study in Table 2.
> We updated Figure 2 as suggested, moreover we added new figures with the interpretable decomposition in the appendix. We exploit the Synthetic datasets proposed in [2] to isolate a specific type of anomaly (e.g. trend or seasonal) and show the behaviour of our decomposition in a controlled experimental setup. Moreover, we added some panels for more complex real world datasets:  Credit Card, Credit Card, GECCO and SWAN-SF datasets (see [2]).
>
> [2] Lai, Kwei-Herng, et al. "Revisiting Time Series Outlier Detection: Definitions and Benchmarks." (2021).
> [3] Robert B. Cleveland, “STL: A Seasonal-Trend Decomposition Procedure Based on Loess”
> [4] Wen, Qingsong, et al. "RobustSTL: A robust seasonal-trend decomposition algorithm for long time series." Proceedings of the AAAI Conference on Artificial Intelligence. Vol. 33. No. 01. 2019.

---

> > ### Comment · Reviewer_rnBY · 2021-11-29
> > **Reply**
> >
> > Thank you for the detailed reply to my concerns. The authors are encouraged to improve the paper with an improved discussion of more recent work and polish the paper from incorporating reviewers' comments, and resubmit to an appropriate venue. I will stay with my original score.

---

### Official Review · Reviewer_TtBt · 2021-11-07

**Correctness:** 3
**Technical Novelty And Significance:** 4
**Empirical Novelty And Significance:** 2
**Recommendation:** 6
**Confidence:** 4

**Main Review:**

Strengths:

* The network architecture (Figure 1) is carefully designed to incorporate useful ideas from both conventional statistical time series models and DNN models to allow the proposed model to get the best of both worlds, including its interpretability and flexibility.

* Although the regularization method for the TCN is inspired by the automatic relevance determination (ARD) method proposed more than two decades ago for (sparse) Bayesian learning of neural networks, its use here is novel as far as I know.

* The nonparametric extension of the classical CUMSUM algorithm for anomaly detection does not require knowledge of the distributions of the two windows of data points, relaxing a major restriction of the original CUMSUM.

Weaknesses:

* Presentation of the experimental results (Table 1) can be improved. For example, some baseline models (e.g., ARIMA) are in fact univariate time series forecasting models. I suppose they handle each dimension of a multivariate time series separately. It would be clearer in the presentation to separate the models into two groups depending on whether they are really multivariate models. In fact, there also exist conventional multivariate time series forecasting models (e.g., VAR). Shouldn’t some of them be included as well? For clarity, it would help to show the dimensionality of each dataset as well, perhaps using a table to summarize the major characteristics (including dimensionality) of all the datasets used.

* While the nonparametric extension of CUMSUM is good, the estimation error can be high (also with high variance) if the windows are not long enough, yet having too long windows may miss some small, subtle anomalies. This is a dilemma. A more detailed analysis is needed.

* The experimental study is a bit limited. It is not clear whether the superior performance of the proposed method shown in Table 1 also holds for a wider range of real-world multivariate time series datasets.

Some general comments:

* You may consider how the proposed automatic complexity determination scheme is related to the attention mechanism.

* For qualitative evaluation, it would help to demonstrate the efficacy of the automatic complexity determination scheme by visualizing the “relevant past” selected automatically.

* In Figure 1, other than the captions of the sub-figures, there should also be a caption for the figure as well. At least it should be shown as “Figure 1” explicitly.

* There are some minor language errors in the paper. It should be proofread carefully to correct them.

**Summary Of The Paper:**

This paper aims to boost the performance of deep neural networks (DNNs) for time series applications by focusing on the characteristics of interpretable forecasting and anomaly detection which are important for real-world time series data. The authors propose an end-to-end trainable DNN architecture which is composed of stacked residual blocks to separate signal components including slow trends, quasi-periodicity, and linear dynamics, followed by a temporal convolutional network (TCN) to model other components. Although previous studies showed that conventional simple linear models often outperformed DNN models on typical time series benchmarks that require robustness and interpretability, this paper shows that the proposed DNN model outperforms state-of-the-art robust statistical methods in some datasets, kind of demonstrating the best of both worlds.


**Summary Of The Review:**

This is a good work which tries to integrate ideas from both conventional time series forecasting models and more recently DNN models. There are some novel ideas which are interesting and are worth studying further. Nevertheless, as mentioned above, the significance of the proposed model for multivariate time series anomaly detection is unknown due to the limitations of the empirical study.

---

> ### Author Response · Authors · 2021-11-22
> **New Benchmark Results**
>
> We sincerely thank the Reviewer for the comments. Below we address all the comments.
>
> ## Benchmarks (reviewer’s point 1 and 4)
> To further address the reviewer’s concerns we tested STRIC on a different anomaly detection benchmark containing real world multivariate time series [1]. In particular our results suggest STRIC outperforms any deep learning based method and any other SOTA alternative in [1] (AR, IForest and OCSVM). Moreover, thanks to our automatic complexity selection, STRIC is less sensitive w.r.t. the length of the window of past values used to feed the time series predictor. Such a hyper-parameter is critical to achieve optimal performance and it is usually selected by cross-validation (very expensive). In these experiments we fixed a common length of 50 past time instants (for all datasets).
>
> In particular STRIC achieves:
> - 5% improvement over GBRT in the Credit Card Dataset
> - 10% improvement over LSTM-RNN in the CICIDS Dataset
> - 7% improvement over IForset in the GECCO Dataset
> - 13% improvement over IForset in the SWAN-SF Dataset
>
> We added these results to Table 1. Moreover, we split Table 1 in univariate and multivariate time series to make the distinction more clear. We point out that to get the results of Table 1 we applied MARIMA models on multivariate time-series and ARIMA on univariate ones, we grouped MARIMA and ARIMA together on the same row for space limitations. This is no longer an issue on the new Table 1.
>
> ## Datasets details (reviewer’s point 2)
> Please see Table 3 for details on the dimension of the datasets. To improve readability we modified this table as done for Table 1 by explicitly showing whether the anomaly detection task is univariate or multivariate.
>
> ## Windows length-variance Trade-off  (reviewer’s point 3)
> We thank the reviewer for pointing out this very important fact. In the updated version of the manuscript we further analyzed the interplay between estimation variance of the likelihood ratios and the windows length.
> In particular our method easily detects persistent changes in the data distribution (change points) by looking at long windows of data, in such cases the estimation variance is small (as observed by the reviewer, and already present in the manuscript). On the other hand, as we look at a smaller time scale, the local sensitivity of our detector to data fluctuations and its estimation variance increases. So that the detector might decrease performance if the abnormal data are not far apart w.r.t. the normal behaviour. We further tested this on our datasets and did not find appreciable changes in detection scores by increasing the windows length (both on Yahoo, NAB and the new benchmark we used [1]). In particular for the Yahoo dataset increasing the detector window length (up to 10 times) does not lead to performance degradation. We believe this is due to the large relative scale of the point anomalies present in the Yahoo benchmark which are easily detected provided a good predictive model is used (see Figure 10 for an example).
> Despite our positive experimental results a possible solution to mitigate the high-variance issue related to small windows is to slightly modify our algorithm and keep more data from the “reference past” so that the estimation variance on the normal behaviour is reduced. Nonetheless this modification improves specificity and not sensitivity (since we do not increase the amount of abnormal data). We leave for future work further studies on sensitivity improvement of our method, we added to the discussion section this relevant future direction.
>
> ## Connection with attention (reviewer’s general comment 1)
> Our automatic complexity selection and the attention mechanism are indeed closely related, since both extract the most relevant information from available data. Nonetheless one important difference is present: our automatic complexity selection is a special form of “structured” attention.  In particular, our automatic complexity selection criterion implicitly assumes the most relevant past is “connected”; by “connected” we mean that it is not possible to have two disjoint important subsequences of past data. On the other hand vanilla attention mechanisms “generalize” our module and allow to consider relevant non-connected subsequences too.
>
> ## More details on the most relevant past (reviewer’s general comment 2)
> This is already present in Figure 3: the relevant past is the point in which the green curve starts being constant. A similar figure is Figure 9. Nonetheless we added a more readable figure in the appendix directly displaying the relevant time scales extracted on different time series.
>
> Thanks for spotting the missing caption, we shall indeed add it to the final version of the manuscript.
>
> [1] Lai, Kwei-Herng, et al. "Revisiting Time Series Outlier Detection: Definitions and Benchmarks." (2021).

---

> > ### Comment · Reviewer_TtBt · 2021-11-30
> > **Thanks for responding to my review**
> >
> > I thank the authors for responding to the comments in my original review. With the changes incorporated in the revised version, it confirms my relatively positive view on the paper.

---

### Official Review · Reviewer_zX4p · 2021-11-07

**Correctness:** 2
**Technical Novelty And Significance:** 2
**Empirical Novelty And Significance:** 2
**Recommendation:** 5
**Confidence:** 4

**Details Of Ethics Concerns:**

N.A.

**Main Review:**

Strengths: The regularization method introduced in Section 4 facilitates avoiding model overfitting.

Weaknesses:

1. Experimental results are not convincing. The chosen baselines are not state-of-the-art solutions for TS anomaly detection, please refer to [1] for details. The benchmark dataset used for comparison should also be changed. For example, as pointed out in [2], Yahoo dataset is not preferred for comparing AD solutions.

2. TS decomposition itself is a well-studied research problem and it is essential to compare the proposed architecture with existing techniques (e.g., [3]) to obtain the residual.

3. The paper is difficult to follow. Firstly, the overall workflow for anomaly detection is not clear, and it is difficult to see the connections between Section 4 and Section 5. Secondly, the model is not described clearly. For example, what is the non-linear module? Is it the TCN block shown in the figure without a title? Thirdly, there does not exist any statement on how the anomaly detector determines a particular time point or subsequence to be an anomaly.

4. The paper is not focused and swings between prediction, anomaly detection, and even text embeddings.


References:

[1] Lai, Kwei-Herng, et al. "Revisiting Time Series Outlier Detection: Definitions and Benchmarks." (2021).

[2] Wu, Renjie, and Eamonn Keogh. "Current time series anomaly detection benchmarks are flawed and are creating the illusion of progress." IEEE Transactions on Knowledge and Data Engineering (2021).

[3] Wen, Qingsong, et al. "RobustSTL: A robust seasonal-trend decomposition algorithm for long time series." Proceedings of the AAAI Conference on Artificial Intelligence. Vol. 33. No. 01. 2019.

**Summary Of The Paper:**

This paper proposed a residual-style architecture namely STRIC for multivariate time series (TS) forecasting and anomaly detection, by first decomposing TS to trend, seasonality, and irregular components as residual, then extracting features from the residual using TCN, and finally introducing a likelihood ratio estimation method. Experimental results show that STRIC can provide interpretability for TS forecasting and outperform some existing anomaly detection models.

**Summary Of The Review:**

While some of the proposed techniques seem to be interesting, the paper is very difficult to follow and the experimental results are not convincing.

---

> ### Author Response · Authors · 2021-11-22
> **New Benchmark Results & discussion on STL-like decompositions**
>
> We sincerely thank the Reviewer for the comments. Below we address all the comments.
>
> ## Benchmarks (point 1)
> While paper [1] will appear in press at the next NeurIPS conference, following the reviewer suggestion, we have evaluated our model on the multivariate benchmark datasets proposed in [1].
>
> Our method (STRIC) outperforms models in [1] for all but one of the 5 synthetic datasets at different contamination rates. Due to lack of space we now report results only @20% contamination, compared to the best-performing model for each synthetic dataset in [1], we obtain:
>
> - 13% better than GBRT for Trend Outliers
> - 9% better than subseq OCSVM for Seasonal Outliers
> - 7% better than AR for Shapelet Outliers
> - 3% better than AR for Contextual Outliers
> - 9%  worse than subseq OCSVM for Global Outliers
>
> For any real world dataset in [1] STRIC outperforms both deep learning based methods and other SOTA alternatives tested in [1]. Moreover, thanks to our automatic complexity selection, STRIC is less sensitive to  the length of the window of past values used as input: we fixed it to 50 for all datasets.
>
> In particular STRIC achieves:
> - 5% better than GBRT in the Credit Card Dataset
> - 10% better than LSTM-RNN in the CICIDS Dataset
> - 7% better than IForset in the GECCO Dataset
> - 13% better than IForset in the SWAN-SF Dataset
>
> Further comments on the experimental results:
> STRIC’s success can be attributed to the fact that it adopts contextual points (the relevant past) to model the data normal behaviour.
> STRIC does not achieve superior performance (while being comparable) on synthetic datasets containing many global outliers. For these datasets the best model class is the subseq OCSVM (similarity based method). This is not surprising since the global outlier points are out of scale points which can simply be isolated by similarity based detection methods even when the available number of data is scarce.
>
> ## Comparison with Robust-STL (point 2)
> The residuals produced by a Robust-STL like decomposition cannot be used to detect change points on trend and seasonal components since they are unaffected by such abnormal behaviour. Robust-STL can easily model jumps in trend and changes in the seasonal pattern. This is clear by visually inspecting Figure 3 (a) in [3]: there is no way to tell that a change point has happened (while many step jumps are present) just by looking at the Robust-STL residual (third panel).
> On the other hand, standard STL decomposition is indeed a valid method since it does not model the abnormal jumps in trend and seasonality. Nonetheless its high sensitivity to noise makes this technique less effective in practice (see next experimental results).
> STRIC extends both STL and Robust-STL decompositions by avoiding the aforementioned limitations.
> To further illustrate our point, we compared the residual extracted by STL and Robust-STL (using the official code of [3]) on the synthetic benchmark proposed in [1]. We could not apply Robust-STL on the real world dataset due to its high computational requirements on long time series (see [3] for more details). Moreover, note that Robust-STL is NOT a multivariate method and cannot describe entity level anomalies (i.e. anomalies detectable only by looking jointly at multiple time series).
> @20% contamination STRIC improves F1 scores on average by 20% w.r.t. STL and by 25% w.r.t. Robust-STL.
>
> # Anomaly detection workflow (points 3.a,b,c,d)
> Our method is an instance of “prediction Deviation method” [1]. To sum up, we tackle the anomaly detection task by building a sufficient statistic: “the prediction error” of a predictive model of the normal behaviour of the time series which then we further analyze using our novel non-parametric estimator.
> This is a well established approach which we already described in the introduction.
> Our main contribution is to propose a novel predictive model specifically designed to have high representational capability (TCN module), decompose time series (interpretable module) and avoid overfitting (fading regularization). Moreover we introduced a novel non parametric estimator of the likelihood ratios (nominal vs. abnormal, see Eq. 3)  which is especially useful in our setup since the residual distribution cannot be assumed a priori and no other standard CUMSUM-based algorithm could be applied.
>
> To improve clarity of our presentation, we:
> - changed section 5’s title to “Anomaly score on prediction residuals”,
> - further highlighted the two stage nature of our method,
> - added a new figure depicting the entire two stage pipeline in Figure 1,
> - moved Figure 1b to appendix and added to it a new panel devoted to the non-linear module,
> - we added all the steps necessary to compute the anomaly scores starting from the data through the predictive model and the non-parametric CUMSUM algorithm.
>
> Last, text embeddings are merely exploited to illustrate the wide applicability of our method to complex high dimensional datasets.

---

> > ### Comment · Reviewer_zX4p · 2021-11-29
> > **Thank you for the responses**
> >
> > The authors have addressed some of my earlier concerns and I have raised the score accordingly.

---

### Decision · Program_Chairs · 2022-01-20

**Decision:**

Reject

**Comment:**

This paper studies the important problem of time series anomaly detection using deep neural networks (DNNs). Unlike many other DNN models, it focuses on incorporating in its model architecture interpretable components that are inspired by previous studies based on both conventional statistical methods and more recent DNN models.

While the paper has merits as pointed out by the reviewers (esp. TtBt), a number of concerns have also been raised, including the choice of datasets (e.g., by reviewers rnBY and zX4p). We appreciate the authors’ effort by adding some preliminary results of further experiments, but addressing all the concerns thoroughly will need a lot more work to get a scholarly paper that is more ready for publication. We believe this work has potential to be accepted for publication in a reputable venue if the concerns are thoroughly addressed after substantial revision.